# Efficient Optimization with Orthogonality Constraint:
# a Randomized Riemannian Submanifold Method

**Andi Han** [1 2]    **Pierre-Louis Poirion** [1]    **Akiko Takeda** [1 3]

## Abstract

Optimization with orthogonality constraints frequently arises in various fields such as machine learning. Riemannian optimization offers a powerful framework for solving these problems by equipping the constraint set with a Riemannian manifold structure and performing optimization intrinsically on the manifold. This approach typically involves computing a search direction in the tangent space and updating variables via a retraction operation. However, as the size of the variables increases, the computational cost of the retraction can become prohibitively high, limiting the applicability of Riemannian optimization to large-scale problems. To address this challenge and enhance scalability, we propose a novel approach that restricts each update on a random submanifold, thereby significantly reducing the per-iteration complexity. We introduce two sampling strategies for selecting the random submanifolds and theoretically analyze the convergence of the proposed methods. We provide convergence results for general nonconvex functions and functions that satisfy Riemannian Polyak–Łojasiewicz condition as well as for stochastic optimization settings. Additionally, we demonstrate how our approach can be generalized to quotient manifolds derived from the orthogonal manifold. Extensive experiments verify the benefits of the proposed method, across a wide variety of problems.

## 1. Introduction

In this paper, we consider optimization problems with orthogonality constraint, i.e.,

$$\min_{X \in \mathbb{R}^{n \times p}: X^\top X = I_p} F(X) \tag{1}$$

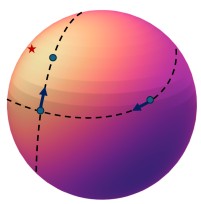

Figure 1: Proposed random submanifold method on 2-sphere. Each iteration restricts the update to a 1-dimensional randomly selected submanifold, i.e., a circle.

where the matrix variable $X \in \mathbb{R}^{n \times p}$ with $n \geq p$ is column orthonormal and $F : \mathbb{R}^{n \times p} \to \mathbb{R}$. Optimization with orthogonality constraint arises naturally in various domains of applications because it is crucial for achieving certain desired properties, such as linear independence, numerical stability and geometry preserving. Examples of applications include principal component analysis (Hotelling, 1933), independent component analysis (Theis et al., 2009), multi-view clustering (Khan & Maji, 2021; Liu et al., 2021; Chen et al., 2022), low-rank matrix completion (Vandereycken, 2013; Mishra et al., 2014), robust optimal transport (Lin et al., 2020; Huang et al., 2021), training of deep neural networks (Helfrich et al., 2018; Li et al., 2019; Wang et al., 2020), continual learning (Chaudhry et al., 2020) and fine-tuning large foundation models (Qiu et al., 2023; Liu et al., 2024), among many others.

Riemannian optimization (Absil et al., 2008; Boumal, 2023; Han et al., 2024b) provides a powerful framework for solving (1) by leveraging the geometry of the orthogonality constraint. Indeed, the set of orthogonal constraint forms a smooth manifold known as the Stiefel manifold, denoted by $\mathrm{St}(n, p) \coloneqq \{X \in \mathbb{R}^{n \times p} : X^\top X = I_p\}$. By equipping the manifold with a suitable Riemannian metric, optimization can be performed intrinsically on the manifold. A crucial step in this process is the retraction operation, which ensures that iterates remain on the manifold after each update. Various retractions have been proposed for the Stiefel manifold, such as those based on QR factorization (Absil et al., 2008), polar decomposition (Absil & Malick, 2012), the Cayley transform (Wen & Yin, 2013), and the matrix exponential (Edelman et al., 1998). All these retractions require non-standard linear algebra operations with a complexity of at least $O(np^2)$ (see Section 2 for details). As a result, the retraction step becomes the primary bottleneck for Riemannian optimization solvers as $n$ and $p$ increase.

---

[1]RIKEN AIP [2]University of Sydney [3]University of Tokyo. Correspondence to: Andi Han <andi.han@sydney.edu.au>.

*Proceedings of the 42$^{nd}$ International Conference on Machine Learning*, Vancouver, Canada. PMLR 267, 2025. Copyright 2025 by the author(s).

In this work, we propose a novel approach that updates the variable on a random submanifold. In particular, our contributions are summarized as follows.

- We propose a novel parameterization for the update via the action of orthogonal group. This allows to update the current iterate in a random submanifold of orthogonal group. This reduces the complexity of non-standard linear algebra operations (such as matrix decomposition, matrix inverse, and matrix exponential) from $O(np^2)$ to $O(r^3)$, where $r$ is the dimension of the submanifold selected.

- We introduce two strategies for the parameterization, through *permutation* and *orthogonal transformation*. We derive the convergence results both in expectation and in high probability. We show the trade-off between the two in terms of efficiency and convergence guarantees. We show the other standard computations are reduced from $O(np^2)$ to $O(nr^2)$ or $O(npr)$ under permutation and orthogonal sampling respectively.

- We establish convergence guarantees for a range of settings, including *general nonconvex* optimization problems, nonconvex functions that satisfy the *Riemannian Polyak-Łojasiewicz (PL)* condition, and *stochastic settings* under both general nonconvex and PL conditions. We show how our developments can be extended to quotient manifolds derived from the orthogonal manifold, including Grassmann and flag manifolds.

- We validate the effectiveness of the proposed method through extensive experiments, showcasing its fast convergence across a variety of problems, spanning both synthetic and real-world applications.

### 1.1. Related Works

Lezcano-Casado & Martınez-Rubio (2019) re-parameterizes the orthogonality constraint in Euclidean space via the Lie exponential map. However, this approach requires differentiating through the exponential map, which can be computationally expensive. Recently, Shalit & Chechik (2014); Gutman & Ho-Nguyen (2023); Yuan (2023); Han et al. (2024a); Cheung et al. (2024) extend the idea of coordinate descent to Stiefel manifold by only updating a few rows/columns while adhering to the orthogonality constraint. Despite the promise in cheap per-iteration update, they either suffer from poor runtime on modern hardware, such as GPUs by requiring a significant number of iterations to converge (Shalit & Chechik, 2014; Gutman & Ho-Nguyen, 2023; Han et al., 2024a) or involve a subproblem that may become difficult to solve in general (Yuan, 2023). It is worth highlighting that Cheung et al. (2024) lift the coordinate updates to the ambient space and then project back to the manifold. Their algorithms still require non-standard linear algebra

operations that cost $O(nr^2)$, where our method scales with $O(r^3)$. As we elucidate the differences to these works in Section 4, our proposed submanifold update includes the coordinate descent as a special case, yet being more efficient in runtime empirically. Another line of research, including (Gao et al., 2019; Xiao et al., 2022; Ablin & Peyré, 2022; Ablin et al., 2023) develop infeasible methods for solving (1), where the updates do not necessarily satisfy the orthogonality constraint. A recent work (Shustin & Avron, 2024) proposes a randomized sketching method on the generalized Stiefel manifold with constraint $X^\top BX = I_p$. However, they assume $B = Z^\top Z$, with $Z \in \mathbb{R}^{d \times n}$ with $d \gg n$. The aim is to reduce complexity in constructing $B$ and improve the conditioning of optimization, which is different to our setting where $B = I_n$ and the aim is to reduce the complexity related to retraction.

## 2. Preliminaries

Stiefel manifold $\mathrm{St}(n,p) = \{X \in \mathbb{R}^{n \times p} : X^\top X = I_p\}$ is the set of column orthonormal matrices. When $n = p$, $\mathrm{St}(n,p) \equiv \mathcal{O}(n)$, which is called orthogonal manifold, also forming a group. We use $T_X \mathrm{St}(n,p)$ to denote the tangent space at $X \in \mathrm{St}(n,p)$ and consider the Euclidean metric as the Riemannian metric, i.e., for any $U, V \in T_X \mathrm{St}(n,p)$, $\langle U, V \rangle_X = \langle U, V \rangle$, we use $\langle \cdot, \cdot \rangle$ to represent the Euclidean inner product. For a smooth function $F : \mathrm{St}(n,p) \to \mathbb{R}$, the Riemannian gradient $\mathrm{grad}F(X) \in T_X \mathrm{St}(n,p)$ is derived as $\mathrm{grad}F(X) = \nabla F(X) - X\{X^\top \nabla F(X)\}_{\mathrm{S}}$, where $\nabla F(X)$ is the Euclidean gradient and $\{A\}_{\mathrm{S}} := (A + A^\top)/2$. Retraction $\mathrm{Retr}_X : T_X \mathrm{St}(n,p) \to \mathrm{St}(n,p)$ is a smooth map that allows to update iterate following a tangent vector direction. Many retractions are proposed on Stiefel manifold, including (1) QR-based retraction: $\mathrm{Retr}_X(U) = \mathrm{qf}(X + U)$, where qf extracts the Q-factor from the QR decomposition; (2) Polar retraction: $\mathrm{Retr}_X(U) = (X + U)(I_p + U^\top U)^{-1/2}$; (3) Cayley retraction: $\mathrm{Retr}_X(U) = (I_n - W)^{-1}(I_n + W)X$ where $U = WX$ for some skew-symmetric $W \in \mathbb{R}^{n \times n}$; (4) Exponential retraction: $\mathrm{Retr}_X(U) = \begin{bmatrix} X & U \end{bmatrix} \mathrm{expm}(\begin{bmatrix} X^\top U & -U^\top U \\ I_p & X^\top U \end{bmatrix}) \begin{bmatrix} \mathrm{expm}(-X^\top U) \\ 0 \end{bmatrix}$, where $\mathrm{expm}(\cdot)$ denotes matrix exponential. We highlight that *all retractions require linear algebra operations other than matrix multiplications that costs at least $O(np^2)$*.

One classic Riemannian solver is the Riemannian gradient descent (Udriste, 2013) that updates the variable as $X_{k+1} = \mathrm{Retr}_{X_k}(-\eta_k \mathrm{grad}F(X_k))$, where $\eta_k > 0$ is the stepsize. Other more advanced solvers include Riemannian accelerated gradient methods (Ahn & Sra, 2020; Alimisis et al., 2021; Han et al., 2023b), variance reduced gradient methods (Bonnabel, 2013; Zhang et al., 2016; Kasai et al., 2018; Han & Gao, 2020; 2021a;b; Utpala et al., 2023), quasi-

Newton methods (Qi et al., 2010; Huang et al., 2015; 2018), and second-order methods (Absil et al., 2007; Agarwal et al., 2021), just to name a few. Some more recent works develop algorithms for solving non-smooth optimization (Ferreira & Oliveira, 1998; Chen et al., 2020; Li et al., 2024), min-max optimization (Zhang et al., 2023; Han et al., 2023a;c), bi-level optimization problems (Han et al., 2024c; Li & Ma, 2025) on Riemannian manifolds.

Despite the recent progress in Riemannian optimization, *All the aforementioned methods utilize the retraction operation* and thus it becomes critical to reduce its complexity before scaling to large problems.

**Notations.** We use $O(\cdot)$ and $\Omega(\cdot)$ to denote the big-O and big-Omega notation and $\mathcal{O}(n)$ to represent the orthogonal manifold of size $n \times n$. $\mathcal{P}(n) \subset \mathcal{O}(n)$ represents the set of permutation matrices and $\mathcal{S}^n = \{v \in \mathbb{R}^n : v^\top v = 1\}$ is the unit sphere. We use $\cong$ to represent a diffeomorphism between two manifolds. We use $\langle \cdot, \cdot \rangle, \| \cdot \|$ to denote the Euclidean inner product and Euclidean norm, and use $\langle \cdot, \cdot \rangle_X, \| \cdot \|_X$ to denote Riemannian inner product and norm on $T_X \mathrm{St}(n, p)$. Because we only consider Euclidean metric as the Riemannian metric in this work, we use $\| \cdot \|$ and $\| \cdot \|_X$ interchangeably. We use $P(r)$ to denote the first $r$ rows of a matrix $P$.

## 3. Riemannian Random Submanifold Descent

This section introduces the proposed method that reduces the complexity of retraction by restricting the update to a random submanifold. In particular, at each iteration $k$, we parameterize the next iterate as $X_{k+1} = U_k X_k$ for some $U_k \in \mathcal{O}(n)$ and thus converts the problem to optimization over $U_k \in \mathcal{O}(n)$ on the orthogonal manifold. Such a parameterization can be justified by the fact that the action of the orthogonal group $\mathcal{O}(n)$ over $\mathrm{St}(n, p)$ is transitive. Hence, at any point $X_k$, there exists a matrix $U_k^* \in \mathcal{O}(n)$ such that $X^* = U_k^* X_k$, where $X^*$ is any local minimizer. Further, we parameterize the orthogonal matrix $U_k$ by a random orthogonal matrix $P_k \in \mathcal{O}(n)$ and a low-dimensional orthogonal matrix $Y \in \mathcal{O}(r)$ where $r$ is the lower dimension that we choose, and we define

$$U_k(Y) = P_k^\top \begin{bmatrix} Y & 0 \\ 0 & I_{n-r} \end{bmatrix} P_k. \qquad (2)$$

By minimizing $\widetilde{F}_k(Y) := F(U_k(Y)X_k)$ over $Y$ instead of minimizing $F_k(U) := F(UX_k)$ over $U$, we update the iterates on a random submanifold defined via (the first $r$ rows of) the random orthogonal matrix $P_k$. Rather than minimizing $\widetilde{F}_k$ to global optimality, we minimize the first-order approximation of the function $\widetilde{F}_k$ around $I_r$ such that the update remains close to $X_k$. This suggests we can compute $Y$ by taking a Riemannian gradient update from $I_r$, i.e., $Y = \mathrm{Retr}_{I_r}(-\eta_k \mathrm{grad}\widetilde{F}_k(I_r))$ for some stepsize $\eta_k > 0$.

---

**Algorithm 1** RSDM

1: Initialize $X_0 \in \mathrm{St}(n, p)$.
2: **for** $k = 0, ..., K - 1$ **do**
3:     Sample $P_k \in \mathcal{O}(n)$ and let $\widetilde{F}_k(Y) = F(U_k(Y)X_k)$ where $U_k(Y)$ is defined in (2).
4:     Compute Riemannian gradient $\mathrm{grad}\widetilde{F}_k(I_r)$.
5:     Update $Y_k = \mathrm{Retr}_{I_r}(-\eta \, \mathrm{grad}\widetilde{F}_k(I_r))$.
6:     Set $X_{k+1} = U_k(Y_k)X_k$.
7: **end for**

---

We remark that our approach can be viewed as a generalization of the random subspace gradient descent in the Euclidean space (Kozak et al., 2021) to the Stiefel manifold. Specifically, the Euclidean random subspace updates the variable as $x_{k+1} = x_k + u_k(y)$ where $u_k(y) = P_k^\top y$ for some random matrix $P_k$ that spans the subspace.

To compute the Riemannian gradient $\mathrm{grad}\widetilde{F}_k(I_r)$, let $P_k(r) \in \mathbb{R}^{r \times n}$ denote the first $r$ rows of $P_k$. Using the expression of Riemannian gradient, we can derive

$$\begin{aligned} \mathrm{grad}\widetilde{F}_k(I_r) &= \frac{1}{2}(\nabla \widetilde{F}_k(I_r) - \nabla \widetilde{F}_k(I_r)^\top) \\ &= \frac{1}{2} P_k(r) \Big( \nabla F(X_k) X_k^\top - X_k \nabla F(X_k)^\top \Big) P_k(r)^\top \\ &= P_k(r) \mathrm{grad} F_k(I_n) P_k(r)^\top \end{aligned} \qquad (3)$$

To ensure the updates adequately explore the full space with high probability, we re-sample the orthogonal matrix $P_k \in \mathcal{O}(n)$ each iteration. The distribution from which the orthogonal matrix is sampled will be discussed in detail in the subsequent section. The full algorithm is outlined in Algorithm 1, where we call the proposed method **Riemannian random submanifold descent (RSDM)**.

## 4. Sampling Strategies and Complexities

In this section, we introduce two sampling strategies and respectively analyze the per-iteration complexity of Algorithm 1. We propose to sample $P_k$ from two distributions, (1) a uniform distribution over the set of *orthogonal matrices* and (2) a uniform distribution over the set of *permutation matrices*. The second strategy of sampling from a permutation matrix is considered due to its sampling and computational efficiency compared to the orthogonal sampling.

The per-iteration complexity of Algorithm 1 is attributed to four parts, i.e., *sampling*, *gradient computation*, *gradient descent update* and *iterate update*. For both sampling strategies, the gradient descent update (Step 5) shares the same complexity. In particular, the gradient update involves the retraction on $\mathcal{O}(r)$, which is on the order of $O(r^3)$. Next we respectively discuss the sampling and computational cost of each sampling strategy. As we show later, the per-iteration cost of orthogonal sampling strategy is $O(npr)$

while the per-iteration cost of permutation sampling strategy is $O(nr^2)$.

**Uniform Orthogonal.** We first consider sampling $P_k$ from the unique translation invariant probability measure (i.e. the Haar measure) on $\mathcal{O}(n)$. Because only $r$ rows of $P_k$ is required, the sampling can be performed using QR decomposition (with Gram-Schmidt method) on a randomly sampled Gaussian matrix $P \in \mathbb{R}^{r \times n}$ (Meckes, 2019). Hence, the cost of sampling is $O(nr^2)$. Furthermore, from (3), the computation of $\mathrm{grad}\widetilde{F}_k(I_r)$ requires a complexity of $O(npr)$ by first computing $P_k(r)\nabla F(X_k)$ and $P_k(r)X_k$ before multiplication. For the iterate update, the computation of $U_k(Y_k)X_k$ only depends on the first $r$ rows of $P_k$ as $U_k(Y_k)X_k = X_k + P_k(r)^\top (Y - I_r)P_k(r)X_k$, which requires $O(npr)$. This suggests the total per-iteration complexity for uniform orthogonal sampling is $O(npr)$.

**Uniform Permutation.** Sampling from a uniform distribution on permutation matrices corresponds to sampling a permutation $\pi : [n] \to [n]$, and thus the sampling complexity is negligible compared to other operations. In practice, sampling $r$ indices without replacement from $[n]$ is sufficient. In terms of gradient computation, because $P_k(r)$ is a truncated permutation matrix, $P_k(r)\nabla F(X_k)$ and $P_k(r)X_k$ corresponds to permuting the rows of $\nabla F(X_k)$ and $X_k$. This largely reduces the cost compared to matrix multiplication. Thus the gradient computation only requires $O(nr^2)$. Lastly, for the iterate update, we highlight matrix multiplication involving both $P_k(r)$ and $P_k(r)^\top$ corresponds to rearranging the rows and thus the cost can be reduced to $O(nr^2)$. Thus the total cost is $O(nr^2)$.

*Remark* 4.1 (**Riemannian coordinate descent is a special case**). We show that with the permutation sampling and $r = 2$, RSDM is equivalent to the Riemannian coordinate descent on Stiefel manifold (Shalit & Chechik, 2014; Han et al., 2024a; Yuan, 2023). To see this, we first recall that a Givens rotation $G_{k,l}(\theta) \in \mathcal{O}(n)$ represents a sparse orthogonal matrix such that its non-zero entries satisfy (1) $[G_{k,l}(\theta)]_{i,i} = 1$ for all $i \neq k$ and $i \neq l$; (2) $[G_{k,l}(\theta)]_{i,i} = \cos\theta$ for all $i = k, l$; (3) $[G_{k,l}(\theta)]_{k,l} = -[G_{k,l}(\theta)]_{l,k} = -\sin\theta$. Further we know that when $r = 2$, any $Y \in \mathcal{O}(2)$ can be parameterized by an angular parameter $\theta$ and is either a rotation matrix $R(\theta) = \begin{bmatrix} \cos\theta & \sin\theta \\ -\sin\theta & \cos\theta \end{bmatrix}$ or a reflection matrix $F(\theta) = \begin{bmatrix} \cos\theta & \sin\theta \\ \sin\theta & -\cos\theta \end{bmatrix}$. Thus it is easy to verify that $G_{k,l}(\theta) = P_{k,l} \begin{bmatrix} R(\theta) & 0 \\ 0 & I_{n-2} \end{bmatrix} P_{k,l}^\top$, where $P_{k,l}$ corresponds to the permutation $\pi$ such that $\pi(1) = k, \pi(2) = l$. This suggests that the update of RSDM reduces to $X_{k+1} = G_{k,l}(\theta)X_k$, which is how coordinate descent is implemented in (Shalit & Chechik, 2014; Han et al., 2024a; Yuan, 2023). In (Yuan, 2023), $F(\theta)$ is further considered as an alternative to the rotation.

## 5. Theoretical Guarantees

In this section, we analyze the convergence guarantees for the proposed RSDM under both orthogonal sampling and permutation sampling. The proofs of all results are included in Appendix sections. We make use of the following notations throughout the section. Recall we have defined in Section 3 that $F_k(U) = F(UX_k)$ and $\widetilde{F}_k(Y) = F(U_k(Y)X_k)$ at iteration $k$. We also introduce generic notations that $F_X(U) := F(UX)$ and $\widetilde{F}_X(Y) = F(U(Y)X)$ for some sampled $P \in \mathcal{O}(n)$.

**Assumption 5.1.** $F$ has bounded gradient and Hessian in the ambient space, i.e., $\|\nabla F(X)\| \leq C_0$, $\|\nabla^2 F(X)[U]\| \leq C_1\|U\|$ for any $X \in \mathrm{St}(n, p), U \in \mathbb{R}^{n \times p}$.

Assumption 5.1 is naturally satisfied given $X \in \mathrm{St}(n, p)$, which is a compact submanifold of the ambient Euclidean space $\mathbb{R}^{n \times p}$. The next lemma verifies the (Riemannian) smoothness of $\widetilde{F}_X(Y)$, which is due to Assumption 5.1.

**Lemma 5.2.** *Under Assumption 5.1, for any $X \in \mathrm{St}(n, p)$, $\widetilde{F}_X(Y)$ is $(C_0 + C_1)$-smooth on $\mathcal{O}(r)$.*

Further, we show the following lemma that relates the gradient of $F_X$ at identity to gradient of $F(X)$.

**Lemma 5.3.** *For any $X \in \mathrm{St}(n, p)$, we can show $\|\mathrm{grad}F_X(I_n)\|^2 \geq \frac{1}{2}\|\mathrm{grad}F(X)\|^2$.*

Apart from general non-convex functions, we also analyze convergence of RSDM under (local) Riemannian Polyak-Łojasiewicz (PL) condition.

**Definition 5.4** (Riemannian Polyak-Łojasiewicz). For a subset $\mathcal{U} \subseteq \mathrm{St}(n, p)$, a smooth function $F : \mathcal{U} \to \mathbb{R}$ satisfies the Riemannian Polyak-Łojasiewicz (PL) condition on $\mathcal{U}$ if there exists $\mu > 0$ such that $\forall X \in \mathcal{U}$, we have $F(X) - \min_{X \in \mathcal{U}} F(X) \leq \frac{1}{2\mu}\|\mathrm{grad}F(X)\|^2$.

### 5.1. Main Results

This section derives theoretical guarantees for the proposed method. A summary of the main results are presented in Table 1. We first give the following proposition that relates the gradient of $\widetilde{F}$ to gradient of $F$ at identity.

**Proposition 5.5.** *Assume that $P$ is uniformly sampled from $\mathcal{P}(n)$ or uniformly sampled from $\mathcal{O}(n)$. Then for any $X \in \mathrm{St}(n, p)$, we have $\mathbb{E}\|\mathrm{grad}\widetilde{F}_X(I_r)\|^2 = \frac{r(r-1)}{n(n-1)}\|\mathrm{grad}F_X(I_n)\|^2$, where the expectation is with respect to the randomness in $P$.*

*Remark* 5.6 (Proof techniques of Proposition 5.5). We obtain the same rate for both the permutation and orthogonal sampling strategies. Nevertheless, the proof techniques are largely different. In the permutation case, the proof boils down to counting the number of permutations that satisfy some criterion and in the orthogonal case, we have to compute, for all set of indices, $\mathbb{E}[P_{ik_1}P_{jl_1}P_{ik_2}P_{jl_2}]$ for $i \neq j$.

Table 1: Summary of the main results under deterministic and stochastic settings, with both orthogonal (ortho.) and permutation (permu.) sampling. The global and local rates refer to the convergence under general nonconvex and PL conditions respectively. Size of Stiefel manifold is $n \times p$ and $k$ is the iteration number. Function constants $L, \mu$ are ignored.

| | | EXPECTATION | | HIGH PROBABILITY | |
| --- | --- | --- | --- | --- | --- |
| | | GLOBAL | LOCAL (PL) | GLOBAL | LOCAL (PL) |
| **Deterministic** (Theorem 5.7, 5.9) | Ortho. | $O\left(\frac{n^2}{r^2 k}\right)$ | $O\left(\exp(-\frac{r^2}{n^2}k)\right)$ | $O(\frac{n^2}{r^2 k})$ | $O\left(\exp(-\frac{r^2}{n^2}k)\right)$ |
| | Permu. | | | $O(\frac{n^2}{r^2 k}\binom{n}{r})$ | $O\left(\exp(-\frac{r^2}{n^2}\binom{n}{r}^{-1}k)\right)$ |
| **Stochastic** (Theorem 5.13) | Ortho. | $O\left(\frac{n^2}{r^2\sqrt{k}}\right)$ | $O\left(\exp(-\frac{r^2}{n^2}k)+\frac{n^2\sigma^2}{r^2}\right)$ | — | |
| | Permu. | | | | |

This is achieved by leveraging the rotational invariance of the distribution of $P$.

Proposition 5.5 shows that the submanifold gradient is on the order of $O(r^2 n^{-2})$ of the full-space gradient. In contrast, the Euclidean subspace gradient method (Kozak et al., 2021) achieves a scaling of $O(rn^{-1})$. This is because our proposed submanifold approach requires applying the projection matrix $P_k$ twice, whereas the Euclidean subspace method requires only a single $P_k$.

### 5.1.1. CONVERGENCE IN EXPECTATION

**Theorem 5.7.** *When $P_k$ is sampled uniformly from $\mathcal{P}(n)$ or $\mathcal{O}(n)$, under Assumption 5.1 and select $\eta = \frac{1}{L}$ with $L = C_0 + C_1$, we obtain that for all $k \geq 1$,*

$$\min_{i=0,\dots,k-1} \mathbb{E}[\|\mathrm{grad}F(X_i)\|^2] \leq \frac{4L\Delta_0}{k}\frac{n(n-1)}{r(r-1)},$$

*where we denote $\Delta_0 = F(X_0) - F^*$. Suppose further $X_k$ converges to a neighborhood $\mathcal{U}$ that contains an (isolated) local minimizer $X^*$. Further, $F$ satisfies Riemannian PL condition on $\mathcal{U}$. Let $k_0$ be that $X_{k_0} \in \mathcal{U}$. Then we have $X_{k_0+k} \in \mathcal{U}, \forall k \geq 1$ and $\mathbb{E}[F(X_{k_0+k})-F(X^*)] \leq \exp\left(-\frac{\mu}{2L}\frac{r(r-1)}{n(n-1)}k\right)\mathbb{E}[F(X_{k_0}) - F(X^*)]$.*

Theorem 5.7 shows that the convergence rate for general non-convex functions maintains the same sublinear convergence, compared with the Riemannian gradient descent (RGD) (Boumal, 2023), albeit with an additional $O(n^2 r^{-2})$ factors. Such a factor can be compensated by the lower per-iteration complexities of RSDM, which leads to a matching total complexity compared to RGD.

*Remark* 5.8 (**Total complexity of RSDM to RGD**). In Theorem 5.7, we show the convergence is at most $O(n^2 r^{-2}/k)$ for both sampling strategies. This implies that in order to reach an $\epsilon$-stationary point in expectation with $\min_{i=0,\dots,k-1}\mathbb{E}[\|\mathrm{grad}F(X_i)\|^2] \leq \epsilon^2$, we require an iteration complexity of $O(n^2 r^{-2}\epsilon^{-2})$, where per-iteration complexity is either $O(npr)$ for orthogonal sampling or $O(nr^2)$ for permutation sampling strategy for permutation sampling, as analyzed in Section 4. This gives a total complexity of at least $O(n^3\epsilon^{-2})$.

This suggests RSDM matches the $O(np^2\epsilon^{-2})$ complexity of Riemannian gradient descent (RGD) in the regime when $p \geq Cn$ for some constant $C > 0$. This corresponds to the challenging regime where the retraction cost dominates other matrix operations. In contrast, when $p \ll n$, the cost of retraction becomes relatively negligible.

### 5.1.2. CONVERGENCE IN HIGH PROBABILITY

Theorem 5.7 suggests both permutation and orthogonal sampling guarantee the same convergence rate in expectation. However, we show in the following theorem that orthogonal sampling achieves much tighter convergence bound in high probability compared to the permutation sampling.

**Theorem 5.9.** *Under Assumption 5.1 and $\eta = \frac{1}{L}$ with $L = C_0 + C_1$, if we use orthogonal sampling, we obtain and for all $k \geq 1$, with probability at least $1 - \exp\left(-\frac{1}{8}(1 - \tau(n,r))k\right)$,*

$$\min_{i=0,\dots,k-1}\|\mathrm{grad}F(X_i)\|^2 \leq \frac{16L\Delta_0}{k}\frac{n(n-1)}{(1-\tau(n,r))r(r-1)}$$

*where $\tau(n,r) = \exp\left(-\frac{r^2(r-1)^2}{2048n^2(n-1)^2}\right)$. If we use permutation sampling, with probability at least $1 - \exp\left(-\frac{1}{8}\binom{n}{r}^{-1}k\right)$,*

$$\min_{i=0,\dots,k-1}\|\mathrm{grad}F(X_i)\|^2 \leq \frac{16L\Delta_0}{k}\frac{n(n-1)}{r(r-1)}\binom{n}{r}$$

*Hence, in both cases, we have that almost surely, $\liminf_{k\to\infty}\|\mathrm{grad}F(X_k)\|^2 = 0$.*

*Under the same setting in Theorem 5.7, suppose $X_{k_0} \in \mathcal{U}$. Then for orthogonal sampling, with probability at least $1 - \exp(-\frac{1}{8}(1 - \tau(n,r))k)$, we have $F(X_{k_0+k}) - F(X^*) \leq \exp\left(-\frac{\mu}{8L}\frac{r(r-1)}{n(n-1)}(1 - \tau(n,r))k\right)\left(F(X_k) - F(X^*)\right)$. For permutation sampling, with probability at least $1 - \exp\left(-\frac{1}{8}\binom{n}{r}^{-1}k\right)$, we have $F(X_{k+1}) - F(X^*) \leq \exp\left(-\frac{\mu}{4L}\frac{r(r-1)}{n(n-1)}\binom{n}{r}^{-1}k\right)\left(F(X_k) - F(X^*)\right)$.*

Theorem 5.9 derives a high-probability bound for for both orthogonal and permutation sampling strategies. For gen-

**Algorithm 2** RSDM with partial deterministic sampling

1: Initialize $X_0 \in \mathrm{St}(n,p)$.
2: **for** $k = 0, ..., K-1$ **do**
3:     Sample $\{P_k^s\}_{s=0}^{S-1}$ such that condition (4) holds.
4:     Set $X_k^0 = X_k$.
5:     **for** $s = 0, ..., S-1$ **do**
6:        Let $\widetilde{F}_k^s(Y) = F(U_k^s(Y)X_k)$ where $U_k^s(Y)$ is defined in (2) with random matrix $P_k^s$.
7:        Compute Riemannian gradient $\mathrm{grad}\widetilde{F}_k^s(I_r)$.
8:        Update $Y_k^s = \mathrm{Retr}_{I_r}(-\eta \, \mathrm{grad}\widetilde{F}_k^s(I_r))$.
9:        Set $X_k^{s+1} = U_k^s(Y_k^s)X_k^s$.
10:     **end for**
11:     Set $X_{k+1} = X_k^S$.
12: **end for**

eral nonconvex functions, it can be seen that the high-probability bound for permutation sampling can be much worse than for the orthogonal sampling due to the additional binomial factor. In addition, compared to orthogonal sampling, permutation sampling requires the number of iteration to be significantly larger in order for the bound to hold with arbitrary probability. To see this, we first can bound $\tau(n,r) \in (0, 0.9995)$ due to $r \leq n$. $0.0005 \leq 1 - \tau(n,r) \leq 1$ and thus $1 - \tau(n,r) = \Theta(1)$. In order to require the high probability bound to hold with probability $1 - \delta'$ (for arbitrary $\delta' \in (0,1)$), we require $k \geq 4000 \log(1/\delta')\delta^{-2} = \widetilde{\Omega}(1)$ for the orthogonal case but require $k \geq 2\binom{n}{r}\log(1/\delta')\delta^{-2} = \widetilde{\Omega}(\binom{n}{r})$, which can be significantly large when $n \gg r$.

*Remark* 5.10 (**The trade-off between efficiency and convergence**). The worse convergence guarantee of permutation sampling relative to orthogonal sampling in high probability indicates a trade-off between efficiency and convergence. Specifically for general nonconvex functions, permutation sampling requires only $O(nr^2)$ complexity per iteration while suffering from a convergence of $O\big(n^2r^{-2}\binom{n}{r}/k\big)$ in high probability. In contrast, orthogonal sampling requires $O(npr)$ complexity per iteration but converges with a rate of $O(n^2r^{-2}/k)$ with high probability. Similar arguments also hold for local linear convergence under PL condition.

## 5.2. Exact Convergence of RSDM

In the previous section, we have derived the convergence rate of RSDM in both expectation and with high probability. Here we adapt RSDM to achieve exact convergence. To this end, we adapt RSDM as described in Algorithm 2. Specifically, we implement RSDM with a double-loop procedure, where for each outer iteration $k$, we sample projection matrices $\{P_k^s\}_{s=0}^{S-1}$ for the $S$ inner iterations, such

that the following non-degenerate condition (4) holds:

$$\sum_{s=0}^{S-1} \|P_k^s(r)\mathrm{grad}F_k(I_n)P_k^s(r)^\top\|^2 \geq C_p\|\mathrm{grad}F_k(I_n)\|^2 \tag{4}$$

Such a non-degenerate condition over the selection of random matrix $P_k^s$ ensures the projected gradient does not vanish. We show in Appendix F that there exist certain sampling schemes that satisfy condition (4).

We now derive the exact convergence guarantees as follows.

**Theorem 5.11.** *Under Assumption 5.1, suppose RSDM is implemented as in Algorithm 2. Then let $\eta = \frac{1}{L}$, with $L = C_0 + C_1$. We can obtain for all $k \geq 1$,*

$$\min_{i=0,...,K-1} \|\mathrm{grad}F(X_i)\|^2 \leq \frac{1}{K}\frac{2L\Delta_0}{C_p(1+C_1^2L^{-2}M^2S^2)},$$

*where $S$ is the inner iteration number, $M$ is a constant depending on the retraction, $C_p$ depends on the sampling.*

## 5.3. Stochastic and Finite-sum Optimization

This section adapts Algorithm 1 for stochastic optimization, defined as

$$\min_{X \in \mathbb{R}^{n \times p}: X^\top X = I_p} \{F(X) \coloneqq \mathbb{E}_\xi[f(X;\xi)]\},$$

where we obtain noisy estimates of the gradients by querying $\xi$. In Algorithm 1, we replace the Riemannian gradient $\mathrm{grad}\widetilde{F}_k(I_r)$ with the stochastic gradient $\mathrm{grad}\tilde{f}_k(I_r;\xi_k)$, where we denote $\tilde{f}(Y;\xi) \coloneqq f(U_k(Y)X_k;\xi_k)$ for some randomly sampled $\xi_k$ at iteration $k$.

For convergence analysis, apart from Assumption 5.1, we also require the assumption of stochastic gradients being unbiased and having bounded variance, which is standard in analyzing stochastic algorithms (Ghadimi & Lan, 2013).

**Assumption 5.12.** The stochastic gradient is unbiased, i.e., $\mathbb{E}_\xi[\nabla f(X;\xi)] = \nabla F(X)$ and has bounded variance, i.e., $\mathbb{E}_\xi[\|\nabla f(X;\xi) - \nabla F(X)\|^2] \leq \sigma^2$, for all $X \in \mathrm{St}(n,p)$.

**Theorem 5.13.** *Under Assumption 5.1 and 5.12, suppose we choose $\eta = \min\{L^{-1}, \sqrt{\Delta_0/L}\sigma^{-1}K^{-1/2}\}$, where we denote $\Delta_0 = F(X_0) - F^*$. Then we can show*

$$\min_{i=0,...K-1} \mathbb{E}\|\mathrm{grad}F(X_k)\|^2 \leq \frac{4n(n-1)}{r(r-1)}\frac{L\Delta_0 + 2\sigma\sqrt{\Delta_0 LK}}{K}$$

*Suppose there exist $X_{k_0}, ..., X_{k_1} \in \mathcal{U}$ for some $k_1 > k_0$, where $\mathcal{U}$ is defined in Theorem 5.7. Then we have $\mathbb{E}[F(X_{k_1}) - F(X^*)] \leq \exp\big(-\frac{\mu}{2L}\frac{r(r-1)}{n(n-1)}(k_1 - k_0)\big)\mathbb{E}[F(X_{k_0}) - F(X^*)] + \frac{\sigma^2}{\mu}\frac{n(n-1)}{r(r-1)}.$*

Theorem 5.13 derives convergence guarantees for stochastic optimization and is comparable to Euclidean analysis under

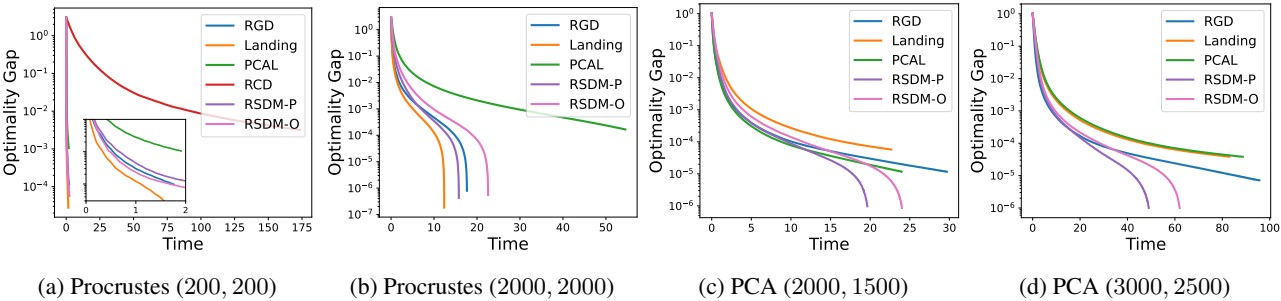

(a) Procrustes (200, 200)    (b) Procrustes (2000, 2000)    (c) PCA (2000, 1500)    (d) PCA (3000, 2500)

Figure 2: Experiments on Procrustes problem and PCA problem under various settings. The numbers in brackets represent the size of $n, p$. For the Procrustes problem, we see RSDM converges competitively against the best baselines due to the simplicity of the problem. For PCA problem, we see RSDM converges the fastest.

general nonconvex functions (Ghadimi & Lan, 2013) and under PL conditions (Garrigos & Gower, 2023). The introduction of additional factor of $O(n^2 r^{-2})$ is consistent with the deterministic setting, analyzed in Section 5.1. Lastly, we remark that although we focus on stochastic gradient in the main paper, the analysis can be easily extended to mini-batch gradient descent, as we show in Appendix E.

### 5.4. Generalization to Quotient Manifolds

We now extend the developments of RSDM to general quotient manifolds. In fact, Stiefel manifold can be equivalently viewed as the quotient space of orthogonal manifold (Edelman et al., 1998). More precisely, we can write $\mathrm{St}(n, p) \cong \mathcal{O}(n)/\mathcal{O}(n-p)$, i.e., a point in the Stiefel manifold corresponds to the equivalence class

$$[Q] = \left\{ Q \begin{pmatrix} I_p & 0 \\ 0 & U \end{pmatrix} : U \in \mathcal{O}(n-p) \right\}.$$

In other words, each point in $\mathrm{St}(n, p)$ is the set of all orthogonal matrices with the same first $p$ columns. Such a viewpoint allows to generalize the previous developments and analysis to more general quotient manifolds of the form

$$\mathcal{M} \cong \mathcal{O}(n)/\mathcal{K} = \{\mathcal{K} \cdot U \, : \, U \in \mathcal{O}(n)\} \quad (5)$$

where $\mathcal{K}$ is a closed subgroup of $\mathcal{O}(n)$. An element of $\mathcal{M}$ is the equivalence class $[Q] = \{K \cdot Q : K \in \mathcal{K}\}$. Quotient manifold of the form (5) includes the famous Grassmann manifold (Edelman et al., 1998), i.e., $\mathrm{Gr}(n, p) \cong \mathcal{O}(n)/(\mathcal{O}(p) \times \mathcal{O}(n-p))$ as well as the flag manifold (Zhu & Shen, 2024), i.e., $\mathrm{Flg}(n_1, \cdots, n_d; n) \cong \mathcal{O}(n)/(\mathcal{O}(n_1) \times \mathcal{O}(n_2 - n_1) \times \cdots \times \mathcal{O}(n_d - n_{d-1}) \times \mathcal{O}(n - n_d))$. Since the action of $\mathcal{O}(n)$ over $\mathcal{M}$ is transitive, we can follow the same approach for $\mathrm{St}(n, p)$, and introduce a function $F_k : \mathcal{O}(n) \mapsto \mathbb{R}$ and $\widetilde{F}_k : \mathcal{O}(r) \mapsto \mathbb{R}$, where $F_k(U) = F(UX_k)$ and $\widetilde{F}_k(Y) = F_k(U_k(Y))$, where $U_k(Y)$ is defined as in (2). We highlight that $X_k \in \mathcal{M}$ is a representation of the equivalence class. For example, in the Grassmann manifold

case, $X_k$ is a column orthonormal matrix whose columns span the subspace. Therefore, Algorithm 1 can be directly applied to the quotient manifolds. Because $F_k$ and $\widetilde{F}_k$ are only defined on the orthogonal manifold, all our results derived for the Stiefel manifold still hold for general quotient manifolds. Apart from the orthogonal group, our developments can also be similarly generalized to other compact matrix groups, such as $SO(n)$.

## 6. Experiments

This section conducts experiments to verify the efficacy of the proposed method. We benchmark our methods with several baseline: (1) Riemannian gradient descent (*RGD*) on Stiefel manifold (Absil et al., 2008; Boumal, 2023); (2) Coordinate descent type of algorithms on Stiefel manifold, namely *RCD* (Han et al., 2024a) and *TSD* (Gutman & Ho-Nguyen, 2023); (3) Infeasible and retraction-free methods, including *PCAL* (Gao et al., 2019) and *Landing* (Ablin & Peyré, 2022; Ablin et al., 2023).

For all experiments, we tune the learning rate in the range of $[0.01, 0.05, 0.1, 0.5, 1.0, 1.5, 2.0]$. For the infeasible methods, we tune the regularization parameter in the range of $[0.1, 0.5, 1.0, 1.5, 2.0]$. For the proposed method (RSDM), we consider both permutation sampling and orthogonal sampling for $P_k$, which we denote as **RSDM-P** and **RSDM-O** respectively. We set the submanifold dimension accordingly based on the problem dimension and fix for both sampling strategies. By defaults, we use QR-based retraction for RGD and proposed RSDM. All experiments are implemented in Pytorch and run on a single RTX4060 GPU. The code is available on https://github.com/andyjm3/RSDM.

### 6.1. Procrustes Problem

We first consider solving the Procrustes problem as to find an orthogonal matrix that aligns two matrices, i.e., $\min_{X \in \mathrm{St}(n,p)} f(X) = \|XA - B\|^2$ for some matrix $A \in$

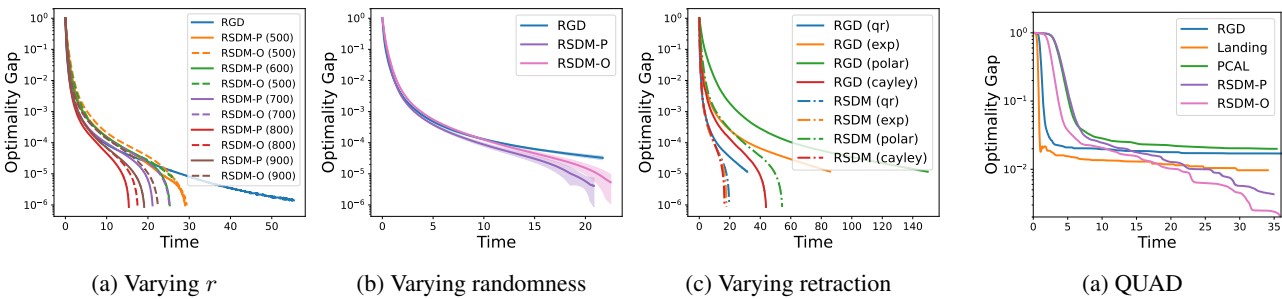

(a) Varying $r$     (b) Varying randomness     (c) Varying retraction     (a) QUAD

Figure 3: Experiment results on PCA ($n = 2000, p = 1500$) by (a) varying low-dimension $r$ and (b) random seed with $r = 700$. The results suggest the outperformance of proposed RSDM over RGD is robust to changes in $r$ as well as random seed.

Figure 4: Experiment results on quadratic assignment problem (QUAD), $n = p = 1000$.

$\mathbb{R}^{p \times p}, B \in \mathbb{R}^{n \times p}$. The optimal solution of this problem is $X^* = UV^\top$ where $U\Sigma V^\top = BA^\top$ is the thin-SVD of the matrix $BA^\top$. Hence the optimal solution is computed as $f(X^*) = \|A\|_F^2 + \|B\|_F^2 - 2\mathrm{tr}(\Sigma)$.

**Setting and Results.** We explore small-size as well as large-size problems by considering (1) $n = p = 200$ and (2) $n = p = 2000$. For the two settings, we set $r = 150$ and $r = 900$ for RSDM respectively. We generate $A, B$ where each entry follows a random Gaussian distribution. For this problem, we compute the closed-form solution $X^*$ by taking SVD of $BA^\top$ and measure the optimality gap in terms of $\mathrm{optgap}(X) = |f(X) - f(X^*)|/|f(X^*)|$. We notice that for feasible methods, like RGD, RCD and proposed RSDM, the problem can be reduced to a linear function as $\max_{X \in \mathrm{St}(n,p)} \langle X, BA^\top \rangle$ while for infeasible methods, the problem remains quadratic. We highlight that because $n = p$, RCD and TSD are equivalent.

In Figure 2(a) under the setting $n = p = 200$, we see RCD performs notably worse compared to other benchmarks in runtime. This is because, although requiring fewer floating point operations (as shown in (Han et al., 2024a)), RCD requires more iterations, which is not GPU-friendly. On the other hand, we see the proposed RSDM performs competitively compared to RGD. When increasing the dimensionality to $n = p = 2000$, we notice RCD requires overly long runtime to progress and thus we remove from the plots. From Figure 2(b), we verify the superiority of RSDM over RGD.

### 6.2. PCA Problem

Next, we consider a quadratic problem, originating from principal component analysis (PCA), as to find the largest eigen-directions of a covariance matrix. This can be formulated as $\min_{X \in \mathrm{St}(n,p)} F(X) = -\mathrm{tr}(X^\top AX)$, where $A \in \mathbb{R}^{n \times n}$. This problem also has analytic solution given by the top-$p$ eigenvectors of $A$.

**Setting and Results.** We create $A$ to be a positive definite

matrix with a condition number of 1000 and exponentially decaying eigenvalues. Due to the existence of an analytic solution, we measure the optimality gap in the same way as in Section 6.1. In Figure 2(c) and (d), we consider the setting of $n = 2000, p = 1500$ and $n = 3000, p = 2500$, which represent large-scale scenarios. For the two settings, we set $r = 700, 1000$ respectively. We see RSDM achieves the fastest convergence among all the baselines. Especially around optimality, we see RSDM switches from the sublinear convergence to linear in contrast to other baselines that maintains the sublinear convergence throughout. This behavior may be attributed to the random projection, which potentially provides a more favorable optimization landscape close to optimum (Fuji et al., 2022). A formal theoretical verification of this claim is left for future work.

To further validate the robustness of RSDM, we conduct additional experiments by varying the low dimension $r$, altering the random seed and utilizing different retractions. The results, presented in Figure 3, demonstrate that the performance of RSDM is largely insensitive to the choice of $r$ (within a reasonable range) and the randomness throughout the iterations. RSDM also consistently outperforms RGD across all available retractions.

### 6.3. Quadratic Assignment Problem

The quadratic assignment problem (Burkard et al., 1997; Wen & Yin, 2013) aims to minimize a quadratic function over permutation matrix. In (Wen & Yin, 2013), the problem is re-formulated as a problem over the Stiefel manifold: $\min_{X \in \mathrm{St}(n,n)} F(X) = \mathrm{tr}(A^\top (X \odot X)B(X \odot X)^\top)$.

**Setting and Results.** We consider the setting of $n = 1000$ and generate $A, B$ as random normal matrices. Since no closed-form solution exists for this problem, we first run RGD for sufficient number of iterations, using the resulting variable as the optimal solution. As shown in Figure 4(a), RSDM converges the fastest among the baselines, especially near the optimal solution. Moreover, in this example,

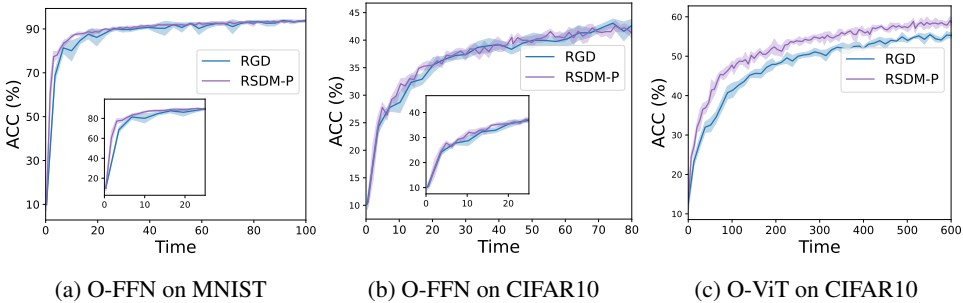

(a) O-FFN on MNIST        (b) O-FFN on CIFAR10        (c) O-ViT on CIFAR10

Figure 5: Test accuracy for training orthogonal neural network (O-FNN) and orthogonal vision transformer (O-ViT) on MNIST dataset and CIFAR10 dataset in five runs.

orthogonal sampling outperforms the permutation sampling.

### 6.4. Orthogonal Neural Networks

We consider optimizing a neural network with orthogonal constraints. We first consider a feedforward neural network (FFN) with ReLU activation for image classification task, i.e., $\min_{\{W_\ell \in \text{St}(d_\ell, d_{\ell+1})\}} L(\text{nn}(X), y)$, where $\text{nn}(X) = \sigma(\cdots \sigma(XW_\ell + b_\ell)\cdots)W_L + b_L$ denotes a L-layer feedforward neural network with bias terms and $L(\cdot, \cdot)$ denotes the cross-entropy loss. Apart from the feedforward neural network, we also consider optimizing an (orthogonal) vision transformer (ViT) (Fei et al., 2022). We follow (Fei et al., 2022) to impose orthogonality constraint on the query, key and value projection matrices.

**Setting and Results.** We optimize neural networks (orthogonal FFN and orthogonal ViT) to classify MNIST (LeCun et al., 1998) and CIFAR10 (Krizhevsky et al., 2009) images. For preprocessing, the MNIST images are resized into $32 \times 32$ for MNIST and CIFAR10 images to $20 \times 20 \times 3$. The images are then normalized into $[-1, 1]$ and vectorized as input for the neural network with a size of 1024 for MNIST and 1200 for CIFAR10. We train a 6-layer FFN, where we constrain the weight of the first 5 layers to be column orthonormal with a hidden size of 1024. The output layer weight, with a size of $1024 \times 10$, remains unconstrained. We also train a 6-layer, 4-head ViT with embedding dimension 1024 and 64 patches, and constrain the query, key and value matrices of all attention layers to be orthogonal. For optimization, we employ RGD and RSDM with a batch size of 16. We set learning rate for unconstrained parameters to be 0.1 and only tune the learning rate for the orthogonal parameters. We plot the test accuracy in Figure 5 where we compare RGD with RSDM-P with five independent runs.

When training an orthogonal FFN, we observe that RSDM achieves faster convergence during the early iterations in terms of runtime, indicating its greater efficiency in quickly reaching high accuracy. On the other hand, when training an orthogonal ViT, RSDM consistently outperforms RGD in test accuracy throughout the training process, with a non-negligible performance gap. This suggests that RSDM may offer greater advantages for training larger and more complex architectures such as ViTs.

## 7. Conclusion

In this paper, we have introduced a novel randomized submanifold approach for optimization problems with orthogonality constraints in order to reduce the high complexity associated with the retraction. We have derived convergence guarantees of the proposed method on a variety of function classes and empirically demonstrated its superiority in a number of problem instances. We also discuss two sampling strategies based on orthogonal and permutation matrices, and discuss the trade-off in terms of computational efficiency versus convergence guarantees.

We believe our developments represent a significant advancement in scalable Riemannian optimization by offering a simple, yet effective solution for large-scale problems with orthogonality constraints. In the paper, we only discuss the application of randomized submanifold strategy to Riemannian gradient descent. Nonetheless, we believe such a strategy can be combined with more advanced optimization techniques, such as line-search (Boumal & Cartis, 2019), momentum (Li et al., 2020; Kong et al., 2023), preconditioning (Kasai et al., 2019) and higher-order methods (Huang et al., 2015; Absil et al., 2007), to further enhance convergence efficiency and robustness with orthogonality constraints and beyond.

**Limitation.** We remark that our random submanifold method has a matching complexity as RGD when $p = \Omega(n)$. Extending our development to the case where $p \ll n$ remains an important future direction.

## Impact Statement

This paper presents work whose goal is to advance the field of Machine Learning. There are many potential societal consequences of our work, none which we feel must be specifically highlighted here.

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

## A. Other related works

Apart from orthogonality constraints, Han et al. (2024a) derive efficient coordinate updates for other matrix manifolds, such as Grassmann and positive definite manifolds. Vary et al. (2024) extend the idea of infeasible update for generalized Stiefel manifold and Darmwal & Rajawat (2023) propose efficient subspace descent algorithms for positive definite manifold with affine-invariance metric. Several other studies (Huang et al., 2021; Peng & Vidal, 2023) investigate (block) coordinate descent on a product of manifolds, where each update targets an individual component manifold. This however is less relevant to our setting, where we exploit submanifolds on a single manifold.

## B. Preliminaries on Stiefel manifold and Riemannian optimization

In this section, we provide more detailed introduction to the Stiefel manifold and Riemannian optimization. Stiefel manifold $\mathrm{St}(n,p) = \{X \in \mathbb{R}^{n \times p} : X^\top X = I_p\}$ is the set of column orthonormal matrices. When $n = p$, $\mathrm{St}(n,p) \equiv \mathcal{O}(n)$, which is called orthogonal manifold, also forming a group. The tangent space of Stiefel manifold is $T_X\mathrm{St}(n,p) = \{U \in \mathbb{R}^{n \times p} : X^\top U + U^\top X = 0\}$. One can choose the Euclidean metric (restricted to the tangent space) as a Riemannian metric for $\mathrm{St}(n,p)$, i.e., for any $X \in \mathrm{St}(n,p)$, and $U, V \in T_X\mathrm{St}(n,p)$, Riemannian metric $\langle U, V \rangle_X = \langle U, V \rangle$ where we use $\langle \cdot, \cdot \rangle$ to represent the Euclidean inner product. Other metrics such as canonical metric (Edelman et al., 1998) can also be considered. Orthogonal projection of any $W \in \mathbb{R}^{n \times p}$ to $T_X\mathrm{St}(n,p)$ with respect to the Euclidean metric is derived as $\mathrm{P}_X(W) = W - X\{X^\top W\}_{\mathrm{S}}$, where we denote $\{A\}_{\mathrm{S}} := (A + A^\top)/2$. For a smooth function $F : \mathrm{St}(n,p) \to \mathbb{R}$, Riemannian gradient of $F$ at $X \in \mathrm{St}(n,p)$, denoted as $\mathrm{grad}F(X)$, is a tangent vector that satisfies for any $U \in T_X\mathrm{St}(n,p)$, $\langle \mathrm{grad}F(X), U \rangle_X = \langle \nabla F(X), U \rangle$, where $\nabla F(X)$ denotes the classic Euclidean gradient. The Riemannian gradient on Stiefel manifold can be computed as $\mathrm{grad}F(X) = \mathrm{P}_X(\nabla F(X)) = \nabla F(X) - X\{X^\top \nabla F(X)\}_{\mathrm{S}}$.

Riemannian optimization works by iteratively updating the variable on the manifold following some descent direction. Throughout the process, a retraction is required to ensure that the iterates stay on the manifold. Specifically, a retraction, denoted as $\mathrm{Retr}_X : T_X\mathrm{St}(n,p) \to \mathrm{St}(n,p)$ is a map from tangent space to the manifold that satisfies $\mathrm{Retr}_X(0) = X$ and $\mathrm{DRetr}_X(0)[V] = V$ for any $V \in T_X\mathrm{St}(n,p)$, where D is the differential operator. There exist various retractions on Stiefel manifold, including (1) QR-based retraction: $\mathrm{Retr}_X(U) = \mathrm{qf}(X + U)$, where qf extracts the Q-factor from the QR decomposition; (2) Polar retraction: $\mathrm{Retr}_X(U) = (X + U)(I_p + U^\top U)^{-1/2}$; (3) Cayley retraction: $\mathrm{Retr}_X(U) = (I_n - W)^{-1}(I_n + W)X$ where $U = WX$ for some skew-symmetric $W \in \mathbb{R}^{n \times n}$; (4) Exponential retraction: $\mathrm{Retr}_X(U) = \begin{bmatrix} X & U \end{bmatrix} \mathrm{expm}(\begin{bmatrix} X^\top U & -U^\top U \\ I_p & X^\top U \end{bmatrix}) \begin{bmatrix} \mathrm{expm}(-X^\top U) \\ 0 \end{bmatrix}$, where $\mathrm{expm}(\cdot)$ denotes matrix exponential. We highlight that all retractions require linear algebra operations other than matrix multiplications that costs at least $O(np^2)$.

## C. Additional experiment results

In the main text, we present the convergence results only in terms of runtime. Here we also plot the convergence with respect to iteration number in Figure 6. We see for Procrustes problem, one of the simplest optimization problems on Stiefel manifold, both RGD and Landing algorithm yields fastest convergence in iteration number. We also notice in small-sized problem, RCD converges quickly. Nonetheless, each iteration of RCD requires to loop through all the $n^2$ indices, resulting in poor parallelizability. This is reflected in the runtime comparisons presented in the main text. For other problem instances, including PCA and quadratic assignment, RSDM attains the fastest convergence not only in runtime (as shown in the main text) but also in terms of iteration count (Figure 6).

Finally, we plot the convergence in iteration for training orthogonal neural networks on MNIST and CIFAR10. We see that RSDM is not able to beat the RGD in terms of convergence in iteration, due to the difficulty of the optimization problems.

## D. Proofs

### D.1. Proof of Lemma 5.2

We first recall the Hessian of a function $G : \mathrm{St}(n,p) \to \mathbb{R}$ along a any tangent vector $V$ is

$$\mathrm{Hess}G(X)[V] = \mathrm{P}_X(\nabla^2 G(X)[V] - V\{X^\top \nabla G(X)\}_{\mathrm{S}})$$

where $\{A\}_{\mathrm{S}} = (A + A^\top)/2$ and $\mathrm{P}_X(\xi) = \xi - X\{X^\top \xi\}_{\mathrm{S}}$.

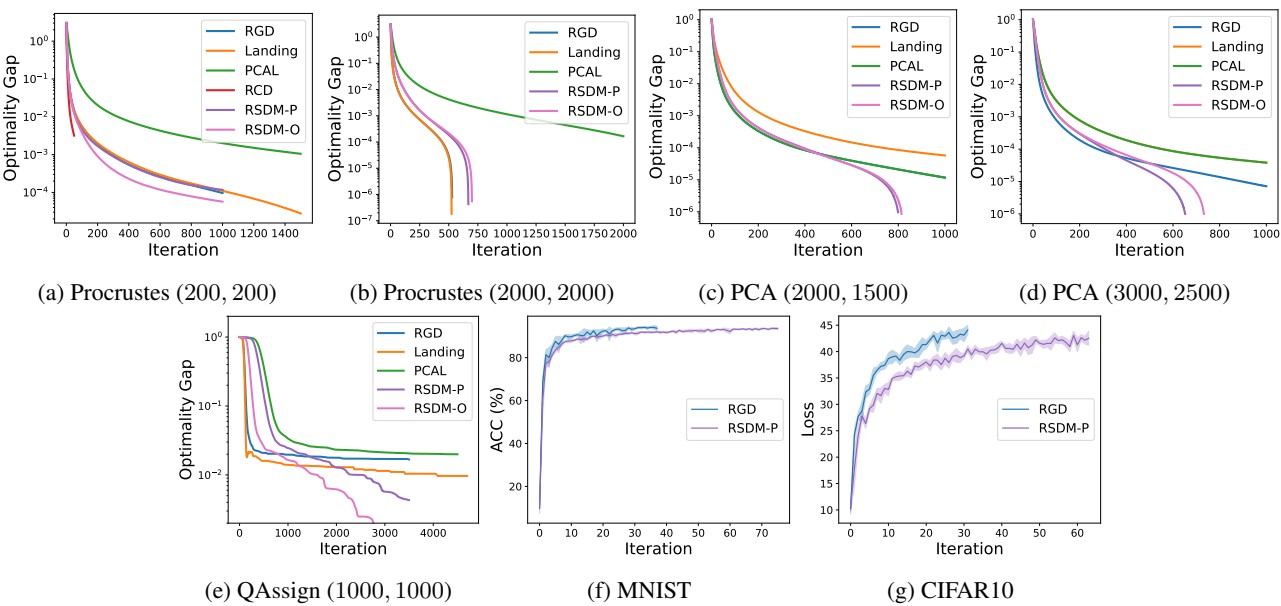

(a) Procrustes (200, 200)    (b) Procrustes (2000, 2000)    (c) PCA (2000, 1500)    (d) PCA (3000, 2500)

(e) QAssign (1000, 1000)     (f) MNIST     (g) CIFAR10

Figure 6: Convergence in terms of iteration on Procrustes problem and PCA problem and quadratic assignment problem under various settings. We observe that except for the Procrustes problem and training of orthogonal neural network, RSDM also converges the fastest in terms of iteration number.

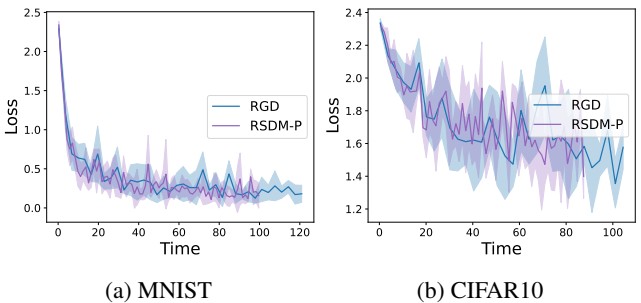

(a) MNIST      (b) CIFAR10

Figure 7: Convergence in loss plot for on image classification.

*Proof of Lemma 5.2.* Recall the (Euclidean) gradient and Hessian of $\widetilde{F}_X(Y)$ is derived as

$$\nabla \widetilde{F}_X(Y) = P(r)\nabla F\big(U(Y)X\big)X^\top P(r)^\top$$
$$\nabla^2 \widetilde{F}_X(Y)[V] = P(r)\nabla^2 F\big(U(Y)X\big)[U(V)X]X^\top P(r)^\top$$

for any $V \in T_Y \mathcal{O}(r)$. This leads to the following Riemannian Hessian

$$\text{Hess}\widetilde{F}_X(Y)[V] = \text{P}_Y\left(\nabla^2 \widetilde{F}_X(Y)[V] - V\{Y^\top \nabla \widetilde{F}_X(Y)\}_\text{S}\right).$$

We wish to bound $\|\text{Hess}\widetilde{F}_X(Y)[V]\|_Y$ in terms of $\|V\|_Y$. First we notice that $\|\text{Hess}\widetilde{F}_X(Y)[V]\|_Y \leq \|\nabla^2 \widetilde{F}_X(Y)[V] -$

$V\{Y^\top \nabla \widetilde{F}_X(Y)\}_{\mathrm{S}}\|$ because for any $\xi$,

$$\begin{aligned}
\|\mathrm{P}_Y(\xi)\|_Y^2 &= \frac{1}{4}\|\xi - Y\xi^\top Y\|^2 = \frac{1}{4}\left(2\|\xi\|^2 - 2\langle \xi, Y\xi^\top Y\rangle\right)\\
&= \frac{1}{4}\left(2\|\xi\|^2 - 2\mathrm{vec}(\xi)^\top (Y^\top \otimes Y)\mathrm{vec}(\xi^\top)\right)\\
&\leq \frac{1}{4}\left(2\|\xi\|^2 + 2\|\xi\|^2\right)\\
&= \|\xi\|^2
\end{aligned}$$

where we use the fact that $\|(Y^\top \otimes Y)v\| = \|v\|$ for any $v$ and $Y \in \mathcal{O}(r)$.

Then we bound

$$\begin{aligned}
\|\nabla^2 \widetilde{F}_X(Y)[V] - V\{Y^\top \nabla \widetilde{F}_X(Y)\}_{\mathrm{S}}\| &\leq \|\nabla^2 \widetilde{F}_X(Y)[V]\| + \|V\|\|\{Y^\top \nabla \widetilde{F}(Y)\}_{\mathrm{S}}\|\\
&\leq \|\nabla^2 F(U(Y)X)[U(V)X]\| + \|V\|\|\nabla F(U(Y)X)\|\\
&\leq C_1\|U(V)\| + C_0\|V\|\\
&= (C_0 + C_1)\|V\|
\end{aligned}$$

where we use triangle inequality in the first inequality. The second inequality uses $\|P(r)\| \leq 1$, $\|X\|, \|Y\| \leq 1$. The third inequality is by assumption on $\nabla^2 F(X), \nabla F(X)$. The last equality is by definition of $U(V)$. $\qquad\square$

### D.2. Proof of Lemma 5.3

*Proof of Lemma 5.3.* From the definition that $F_X(U) = F(UX)$, we let

$$A := \|\mathrm{grad} F_X(I_n)\|^2 = \frac{1}{4}\|\nabla F(X)X^\top - X\nabla F(X)^\top\|^2$$
$$B := \|\mathrm{grad} F(X)\|^2 = \|\nabla F(X) - X\{X^\top \nabla F(X)\}_{\mathrm{S}}\|^2$$

We first notice that

$$\begin{aligned}
A &= \frac{1}{4}\left(\|\nabla F(X)X^\top\|^2 + \|X\nabla F(X)^\top\|^2 - 2\mathrm{tr}(X\nabla F(X)^\top X\nabla F(X)^\top)\right)\\
&= \frac{1}{2}\left(\|\nabla F(X)\|^2 - \mathrm{tr}(X\nabla F(X)^\top X\nabla F(X)^\top)\right).
\end{aligned}$$

Similarly,

$$\begin{aligned}
B &= \|\nabla F(X)\|^2 + \|X\{X^\top \nabla F(X)\}_{\mathrm{S}}\|^2 - 2\mathrm{tr}(\nabla F(X)^\top X\{X^\top \nabla F(X)\}_{\mathrm{S}})\\
&= \|\nabla F(X)\|^2 + \|X\{X^\top \nabla F(X)\}_{\mathrm{S}}\|^2 - \mathrm{tr}(\nabla F(X)^\top X(X^\top \nabla F(X) + \nabla F(X)^\top X))\\
&= \|\nabla F(X)\|^2 - \mathrm{tr}(X\nabla F(X)^\top X\nabla F(X)^\top) + C,
\end{aligned}$$

where we let $C := \|X\{X^\top \nabla F(X)\}_{\mathrm{S}}\|^2 - \mathrm{tr}(\nabla F(X)^\top XX^\top \nabla F(X))$. Then we have

$$\begin{aligned}
C &= \frac{1}{4}\|XX^\top \nabla F(X)\|^2 + \frac{1}{4}\|X\nabla F(X)^\top X\|^2 + \frac{1}{2}\mathrm{tr}(\nabla F(X)^\top X\nabla F(X)^\top X)\\
&\quad - \mathrm{tr}(\nabla F(X)^\top XX^\top \nabla F(X))\\
&= \frac{1}{4}\mathrm{tr}(\nabla F(X)^\top XX^\top \nabla F(X)) + \frac{1}{4}\mathrm{tr}(X^\top \nabla F(X)\nabla F(X)^\top X) + \frac{1}{2}\mathrm{tr}(\nabla F(X)^\top X\nabla F(X)^\top X)\\
&\quad - \mathrm{tr}(\nabla F(X)^\top XX^\top \nabla F(X))\\
&= -\frac{1}{2}\mathrm{tr}(\nabla F(X)\nabla F(X)^\top XX^\top) + \frac{1}{2}\mathrm{tr}(\nabla F(X)^\top X\nabla F(X)^\top X)
\end{aligned}$$

Therefore,

$$B = \|\nabla F(X)\|^2 - \frac{1}{2}\mathrm{tr}(\nabla F(X)^\top XX^\top \nabla F(X)) - \frac{1}{2}\mathrm{tr}(\nabla F(X)^\top X\nabla F(X)^\top X).$$

and,

$$A - B = \frac{1}{2}(\text{tr}(\nabla F(X)^\top X X^\top \nabla F(X)) - \|\nabla F(X)\|^2).$$

We will now prove that

$$A - B \geq -A.$$

To this end, we consider the quantity $\|F(X)^\top X - X^\top F(X)\|^2 \geq 0$. By expanding we obtain

$$
\begin{aligned}
\|F(X)^\top X - X^\top F(X)\|^2 &= \|F(X)^\top X\|^2 + \|X^\top F(X)\|^2 - 2\text{tr}(F(X)^\top X F(X)^\top X) \\
&= 2(\|F(X)^\top X\|^2 - \text{tr}(X F(X)^\top X F(X)^\top)) \\
&= 2(\text{tr}(\nabla F(X)^\top X X^\top \nabla F(X) - \text{tr}(X F(X)^\top X F(X)^\top)) \geq 0.
\end{aligned}
$$

This shows

$$\text{tr}(X \nabla F(X)^\top X \nabla F(X)^\top) \leq \text{tr}(\nabla F(X)^\top X X^\top \nabla F(X)),$$

which concludes $A - B \geq -A$ and thus $A \geq B/2$. $\qquad\square$

### D.3. Proof of Proposition 5.5

*Proof of Proposition 5.5.* We separately prove the results for the strategies of permutation sampling and orthogonal sampling.

**Permutation sampling.** Recall that the gradient $\text{grad}F_X(I_n)$ on $\mathcal{O}(n)$ is given by

$$\text{grad}F_X(I_n) = \frac{1}{2}(\nabla F(X)X^\top - X\nabla F(X)^\top).$$

Hence by (3) and using the fact that $P$ is a permutation matrix, we have

$$\text{grad}\widetilde{F}_X(I_r) = P(r)\text{grad}F_X(I_n)P(r)^\top,$$

where $P(r) \in \mathbb{R}^{r \times n}$ consist of the first $r$ rows of $P$. Let us define by $S_n^r$ the set of all truncated permutation matrix $P(r) \in \mathbb{R}^{r \times n}$. To each element $P$ of $S_n^r$, we can associate a unique "truncated" permutation $\pi$ defined by

$$\forall i \leq r : \ \pi(i) \text{ is such that } P^\top e_i = e_{\pi(i)}.$$

Notice that $\pi$ is defined only for the first $r$ integers as the matrix $P \in S_n^r$ has only $r$ rows. Using the fact that $\text{grad}F_X(I_n)$ is Skew-symmetric, we have

$$\|P(r)\text{grad}F_X(I_n)P(r)^\top\|^2 = 2\sum_{1 \leq i < j \leq r}(\text{grad}F_X(I_n))^2_{(\pi(i),\pi(j))} = \sum_{1 \leq i,j \leq r}(\text{grad}F_X(I_n))^2_{(\pi(i),\pi(j))}.$$

We can check the number of element in $S_n^r$ is given by

$$|S_n^r| = \frac{n!}{(n-r)!},$$

as $n!$ is the number of permutations on $\{1, \cdots, n\}$ and $(n-r)!$ is the number of way to complete the truncated permutation into a permutation on $\{1, \cdots, n\}$. Therefore, we deduce that

$$\mathbb{E}\|\text{grad}\widetilde{F}_X(I_r)\|^2 = 2\frac{(n-r)!}{n!}\sum_{\pi \in S_n^r}\sum_{i<j}(\text{grad}F_X(I_n))^2_{(\pi(i),\pi(j))}.$$

Let us now fix $1 \leq k, l \leq n$. We will now count how many times does the term $(\text{grad}F_X(I_n))^2_{(k,l)}$ appears in the sum

$$\sum_{\pi \in S_n^r}\sum_{i<j}(\text{grad}F_X(I_n))^2_{(\pi(i),\pi(j))}.$$

More formally, let us denote by $n_{k,l} \in \mathbb{N}$, the number of time $(\mathrm{grad}F_X(I_n))^2_{(k,l)}$ appears in the above sum. Then we have that

$$\mathbb{E}\|\mathrm{grad}\widetilde{F}_X(I_r)\|^2 = 2\frac{(n-r)!}{n!} \sum_{1 \le k,l \le n} n_{k,l}(\mathrm{grad}F_X(I_n))^2_{(k,l)}.$$

Let us now compute $n_{k,l}$. For that it is equivalent to count how the number of elements $\pi \in S^r_n$, such that $(k,l) \in \{(\pi(i),\pi(j)) \mid i < j\}$. This equivalent to choosing an ordered pair $(i,j)$ inside $\{1,\cdots,r\}^2$ and then count the number of elements $\pi$ in $S^r_n$ such that $(k,l) = (\pi(i),\pi(j))$. Hence, we deduce that

$$n_{k,l} = \frac{r(r-1)}{2}\frac{(n-2)!}{(n-r)!},$$

as there is exactly $(n-2)!$ permutation $\pi$ such that for fixed $(i,j)$, $(k,l) = (\pi(i),\pi(j))$. We deduce therefore that

$$\mathbb{E}\|\mathrm{grad}\widetilde{F}_X(I_r)\|^2 = 2\frac{(n-r)!}{n!}\frac{r(r-1)}{2}\frac{(n-2)!}{(n-r)!} \sum_{1 \le k,l \le n} (\mathrm{grad}F_X(I_n))^2_{(k,l)}$$

$$= \frac{r(r-1)}{n(n-1)}\|\mathrm{grad}F_X(I_n)\|^2,$$

which completes the proof.

**Orthogonal sampling.** Recall

$$\mathrm{grad}\widetilde{F}_X(I_r) = P(r)\mathrm{grad}F(I_n)P(r)^\top,$$

where $P(r) \in \mathbb{R}^{r \times n}$ consist of the first $r$ rows of $P$. Let us denote by $M := \mathrm{grad}F_X(I_n)$. We have

$$\|\mathrm{grad}\widetilde{F}_X(I_r)\|^2 = \sum_{i,j=1}^r \left(\sum_{k,l=1}^n P_{ik}M_{kl}P_{jl}\right)^2$$

$$= \sum_{i,j=1}^r \sum_{k_1,l_1,k_2,l_2=1}^n P_{ik_1}M_{k_1l_1}P_{jl_1}P_{ik_2}M_{k_2l_2}P_{jl_2}$$

$$= \sum_{i \ne j=1}^r \sum_{k_1,l_1,k_2,l_2=1}^n P_{ik_1}P_{jl_1}P_{ik_2}P_{jl_2}M_{k_1l_1}M_{k_2l_2}$$

where the last inequality holds as $PMP^\top$ is skew symmetric as $M$ is, hence $(PMP^\top)_{ii} = 0$ for all $i$. Hence, to prove the theorem, we must compute, for all set of indices, $\mathbb{E}[P_{ik_1}P_{jl_1}P_{ik_2}P_{jl_2}]$ for $i \ne j$.

First, let us consider the case where all indices $k_1, l_1, k_2, l_2$ are different. We will prove that in such case $\mathbb{E}[P_{ik_1}P_{jl_1}P_{ik_2}P_{jl_2}] = 0$. Indeed, notice that since all the four indices are different, $P_{ik_1}, P_{jl_1}, P_{ik_2}, P_{jl_2}$ belongs to four different columns of $P$. Hence, by multiplying $P$ on the left by an identity matrix where the $1$ are $k_1$ position on the diagonal has been replaced by $-1$, we can change $P_{ik_1}$ to $-P_{ik_1}$ and $P_{ik_1}P_{jl_1}P_{ik_2}P_{jl_2}$ to $-P_{ik_1}P_{jl_1}P_{ik_2}P_{jl_2}$. Since the distribution of $P$ is invariant with such operation, we deduce, by symmetry, that $\mathbb{E}[P_{ik_1}P_{jl_1}P_{ik_2}P_{jl_2}] = 0$.

More generally, let us now consider the case where $k_1, l_1, k_2, l_2$ take at least 3 different values. Then by a similar reasoning, since once column of $P$ must contain at least a single index among $k_1, k_2, l_1, l_2$ then we can show (by multiplying this column by $-1$) that $P_{ik_1}P_{jl_1}P_{ik_2}P_{jl_2}$ has the same distribution as $-P_{ik_1}P_{jl_1}P_{ik_2}P_{jl_2}$, proving again that $\mathbb{E}[P_{ik_1}P_{jl_1}P_{ik_2}P_{jl_2}] = 0$. Hence, we need to consider two cases: $k_1 = k_2$, or $k_1 = l_1$ (notice that $k_1 = l_2$ is the same case as $k_1 = l_1$).

First, let us assume that $k_1 = k_2$. Then, by the previous point, we must also have that $l_1 = l_2$. Now, let us consider the case $k_1 = l_1$. Again, we must have $k_2 = l_2$, otherwise we would have at least 3 distinct columns.

In summary, we have proved, by considering the three cases: $k_1 = k_2, l_1 = l_2$; $k_1 = l_1, k_2 = l_2$ ; and $k_1 = l_2, k_2 = l_1$, that:

$$\mathbb{E}\left[\|\mathrm{grad}\widetilde{F}_X(I_r)\|^2\right]$$

$$= \sum_{i \ne j=1}^r \left(\sum_{l,k=1}^n \mathbb{E}[P^2_{ik}P^2_{jl}]M^2_{kl} + \sum_{l,k=1}^n \mathbb{E}[P_{ik}P_{jk}P_{il}P_{jl}]M_{kk}M_{ll} + \sum_{l,k=1}^n \mathbb{E}[P_{ik}P_{jk}P_{il}P_{jl}]M_{kl}M_{lk}\right).$$

Which implies, by anti-symmetry of $M$ ($M_{lk} = -M_{kl}$ and $M_{kk} = M_{ll} = 0$):

$$\mathbb{E}\left[\|\text{grad}\widetilde{F}_X(I_r)\|^2\right] = \sum_{i\neq j=1}^{r}\sum_{l\neq k=1}^{n}\left(\mathbb{E}[P_{ik}^2 P_{jl}^2] - \mathbb{E}[P_{ik}P_{jk}P_{il}P_{jl}]\right)M_{kl}^2. \tag{6}$$

Let us now compute $\mathbb{E}[P_{ik}^2 P_{jl}^2]$, for $k \neq l$. Notice for all $i$, $\sum_{k=1}^n P_{ik}^2 = 1$. Hence, by multiplying two of this equality (for $i$ and $j$) and taking the expectation, we get that for all $i \neq j$,

$$\sum_{l,k=1}^{n}\mathbb{E}[P_{ik}^2 P_{jl}^2] = 1,$$

which implies that

$$\sum_{l\neq k=1}^{n}\mathbb{E}[P_{ik}^2 P_{jl}^2] = 1 - \sum_{k=1}^{n}\mathbb{E}[P_{ik}^2 P_{jk}^2].$$

However, notice that for all $k \neq l$, the joint law of $P_{ik}, P_{jl}$ is the same. Indeed the law of $P$ does not change by permuting the columns of $P$, which implies that for all $k, l$, the law of $P_{ik}, P_{jl}$ is the same as the joint law of $P_{i1}, P_{j2}$. Hence we have that from the previous equation that

$$(n^2 - n)\mathbb{E}[P_{i1}^2 P_{j2}^2] = 1 - n\mathbb{E}[P_{i1}^2 P_{j1}^2] \tag{7}$$

Furthermore using that

$$\left(\sum_{k=1}^{n}P_{ik}P_{jk}\right)^2 = 0,$$

leading to

$$\sum_{k\neq l=1}^{n}P_{ik}P_{jk}P_{il}P_{jl} + \sum_{k=1}^{n}P_{ik}^2 P_{jk}^2 = 0.$$

Since, by permuting the rows of $P$, $P_{ik}P_{jk}P_{il}P_{jl}$ has the same law as $P_{i1}P_{j1}P_{i2}P_{j2}$, we found that

$$(n^2 - n)\mathbb{E}[P_{i1}P_{j1}P_{i2}P_{j2}] + n\mathbb{E}[P_{i1}^2 P_{j1}^2] = 0. \tag{8}$$

Notice that (6) leads to

$$\mathbb{E}\left[\|\text{grad}\widetilde{F}_X(I_r)\|^2\right] = \sum_{i\neq j=1}^{r}\sum_{l\neq k=1}^{n}\left(\mathbb{E}[P_{i1}^2 P_{j2}^2] - \mathbb{E}[P_{i1}P_{j1}P_{i2}P_{j2}]\right)M_{kl}^2.$$

Hence

$$\mathbb{E}\left[\|\text{grad}\widetilde{F}_X(I_r)\|^2\right] = (r^2 - r)\left(\mathbb{E}[P_{i1}^2 P_{j2}^2] + \mathbb{E}[P_{i1}P_{j1}P_{i2}P_{j2}]\right)\|\text{grad}F_X(I_n)\|^2. \tag{9}$$

Using (7), we found that

$$\mathbb{E}[P_{i1}^2 P_{j2}^2] = \frac{1 - n\mathbb{E}[P_{i1}^2 P_{j1}^2]}{n^2 - n},$$

and using (8), we found that

$$\mathbb{E}[P_{i1}P_{j1}P_{i2}P_{j2}] = -\frac{n\mathbb{E}[P_{i1}^2 P_{j1}^2]}{n^2 - n}$$

Hence,

$$\mathbb{E}\left[\|\text{grad}\widetilde{F}_X(I_r)\|^2\right] = \frac{r^2 - r}{n^2 - n}\|\text{grad}F_X(I_n)\|^2.$$

Thus the proof is now complete. $\qquad\qquad\square$

### D.4. Proof of Theorem 5.9

Here we first prove Theorem 5.9, which is the high probability convergence guarantees. Towards this end, we require the following proposition that deduces a concentration inequality on $\|\mathrm{grad}\widetilde{F}_X(I_r)\|^2$.

**Proposition D.1.** *When $P$ is sampled uniformly from $\mathcal{O}(n)$, we have that*

$$\mathbb{P}\left(\|\mathrm{grad}\widetilde{F}_X(I_r)\|^2 \geq \frac{r(r-1)}{2n(n-1)}\|\mathrm{grad}F_X(I_n)\|^2\right) \geq 1 - \exp\left(-\frac{r^2(r-1)^2}{2048n^2(n-1)^2}\right)$$

*When $P$ is sampled uniformly from $\mathcal{P}(n)$ we have that*

$$\mathbb{P}\left(\|\mathrm{grad}\widetilde{F}_X(I_r)\|^2 \geq \frac{r(r-1)}{n(n-1)}\|\mathrm{grad}F_X(I_n)\|^2\right) \geq \binom{n}{r}^{-1}$$

*Proof of Proposition D.1.* First, let us consider the case where $P$ is sampled uniformly from $\mathcal{O}(n)$. When $P$ is sampled uniformly from $\mathcal{O}(n)$, we can see $P(r)$ is sampled uniformly from $\mathrm{St}(n,r)$. Then by Proposition 5.5,

$$\mathbb{E}\|\mathrm{grad}\widetilde{F}_X(I_r)\|^2 = \frac{r(r-1)}{n(n-1)}\|\mathrm{grad}F_X(I_n)\|^2.$$

In order to derive a high-probability result, we define the following function $h : \mathrm{St}(n,r) \to \mathbb{R}$ that $h(X) = \|X^\top M X\|^2$ where $M \in \mathbb{R}^n$ is any Skew-symmetric matrix. And it can be verified that when $M = \mathrm{grad}F_X(I_n)$, we can show $h(P(r)^\top) = \|\mathrm{grad}\widetilde{F}_X(I_r)\|^2$.

Let us now compute a Lipschitz constant $L_h$ for the function $h$. For that, we compute the Riemannian gradient $\mathrm{grad}h(X)$. Le us first compute the Euclidean gradient $\nabla h(X)$. We have, by anti-symmetry of $M$:

$$\nabla h(X) = -4MXX^\top MX.$$

We therefore deduce the Riemannian gradient:

$$\begin{aligned}\mathrm{grad}h(X) &= -4MXX^\top MX + 4X\{X^\top MXX^\top MX\}_S \\ &= -4MXX^\top MX + 4XX^\top MXX^\top MX\end{aligned}$$

This implies that in order to find the Lipschitz constant $L_h$, we need to bound $\|MXX^\top MX\|$ and $\|XX^\top MXX^\top MX\|$. Using that for any matrix $A, B$, we have that $\|AB\|_F \leq \|A\|_2\|B\|_F$ and that $\|X\|_2 \leq 1$ for any $X \in \mathrm{St}(n,r)$, we can bound the two term above by $\|M\|_F^2$. Hence, we deduce that we can take $L_h = 8\|M\|^2$ as the Lipschitz constant for $h$. From (Götze & Sambale, 2023), we deduce that for any $t > 0$, we have

$$\mathbb{P}\left(h(X) \leq \frac{r(r-1)}{n(n-1)}\|M\|^2 - t\right) \leq \exp\left(-\frac{(n-1)t^2}{512\|M\|^4}\right).$$

Hence, by taking $t = \frac{1}{2}\frac{r(r-1)}{n(n-1)}\|M\|^2$, we deduce that

$$\mathbb{P}\left(h(X) \leq \frac{r(r-1)}{2n(n-1)}\|M\|^2\right) \leq \exp\left(-\frac{r^2(r-1)^2}{2048n^2(n-1)^2}\right).$$

This ends the proof for the case where $P$ is sampled uniformly from $\mathcal{O}(n)$. Notice that the permutation case is obvious as each element is sampled with probability $\binom{n}{r}^{-1}$, and at least one element $P$ should induce a value $h(P(r)^\top)$ larger than $\mathbb{E}[h(P(r)^\top)]$. $\qquad\square$

*Proof of Theorem 5.9.* By Lemma 5.2, we see $\widetilde{F}_X(Y)$ is $L$-smooth with $L = C_0 + C_1$. Then we have for any $Y$ and $W \in T_Y\mathcal{O}(r,r)$,

$$\widetilde{F}_X(\mathrm{Retr}_Y(W)) \leq \widetilde{F}_X(Y) + \langle\mathrm{grad}\widetilde{F}_X(Y), W\rangle + \frac{L}{2}\|W\|^2.$$

Applying this inequality to the update and recalling $\widetilde{F}_k(Y) = F(U(Y)X_k)$, we have

$$F(X_{k+1}) = \widetilde{F}_k(\text{Retr}_{I_r}(-\eta\,\text{grad}\widetilde{F}(I_r))) \le \widetilde{F}_k(I_r) - \left(\eta - \frac{\eta^2 L}{2}\right)\|\text{grad}\widetilde{F}_k(I_r)\|^2$$

$$= F(X_k) - \frac{1}{2L}\|\text{grad}\widetilde{F}_k(I_r)\|^2 \tag{10}$$

where we note that $\widetilde{F}_k(I_r) = F(X_k)$.

Hence, we deduce from Proposition D.1 and Lemma 5.3, that

$$F(X_{k+1}) - F(X_k) \le -\frac{1}{4L}\frac{r(r-1)}{n(n-1)}\|\text{grad}F_k(I_n)\|^2 \le -\frac{1}{8L}\frac{r(r-1)}{n(n-1)}\|\text{grad}F(X_k)\|^2,$$

holds with probability at least $1 - \exp\left(-\frac{r^2(r-1)^2}{2048n^2(n-1)^2}\right) := 1 - \tau(n,r)$. Let us denote, for all $k$, by $Y_k \in \{0,1\}$ the random variable equal to one if and only if the above inequality holds. We have that $\mathbb{E}[Y_k] \ge 1 - \tau(n,r)$, furthermore since $F(X_{k+1}) \le F(x_k)$, we have that for all $k$,

$$\|\text{grad}F(X_k)\|^2 Y_k \le 8L\frac{n(n-1)}{r(r-1)}(F(X_k) - F(X_{k+1})).$$

Hence

$$\left(\min_{i=0,\ldots,k-1}\|\text{grad}F(X_i)\|^2\right)\frac{1}{k}\sum_{i=0}^{k-1}Y_i \le \frac{1}{k}\sum_{i=0}^{k-1}\|\text{grad}F(X_i)\|^2 Y_i \le \frac{8L}{k}\frac{n(n-1)}{r(r-1)}(F(X_0) - F^*).$$

We have by a Chernoff bound (see (Vershynin, 2018)), that for all $\delta \in (0,1)$,

$$\mathbb{P}\left(\sum_{i=0}^{k-1}Y_i \ge (1-\delta)(1-\tau(n,r))k\right) \ge 1 - \exp\left(-\frac{\delta^2}{2}(1-\tau(n,r))k\right). \tag{11}$$

Hence, we deduce that with probability at least $1 - \exp\left(-\frac{\delta^2}{2}(1-\tau(n,r))k\right)$,

$$\min_{i=0,\ldots,k-1}\|\text{grad}F(X_i)\|^2 \le \frac{8L}{k}\frac{n(n-1)}{(1-\delta)(1-\tau(n,r))r(r-1)}(F(X_0) - F^*).$$

Finally, the proof for the permutation case is exactly similar and thus we omit here. This suggests $\liminf_{k\to\infty}\|\text{grad}F(X_k)\|^2 = 0$ almost surely.

For the analysis under the Riemannian PL condition, once there exists $k_0$ such that $X_{k_0} \in \mathcal{U}$, then by the convergence almost surely in gradient norm and the fact that $X^*$ is an isolated local minima, we conclude that there exists a subsequence $X_{k_j}$, for $k_0 \le k_1 < k_2 < \ldots$ converging to $X^*$. For such a subsequence, it is clear that $F(X_{k_j})$ converges to $F(X^*)$. Furthermore, because $F(X_k)$ converges as in (10), then $F(X_k)$ must converge to $F(X^*)$ almost surely. By (Rebjock & Boumal, 2024), we also know that quadratic growth holds (due to PL condition), i.e., $F(X_k) - F(X^*) \ge \frac{\mu}{2}\|X_k - X^*\|^2$ (by $X^*$ is isolated). Then we have $\|X_k - X^*\|^2 \to 0$ almost surely. Thus, $X_k \in \mathcal{U}$ for all $k \ge k_0$.

Next, we derive the convergence rate. If we use orthogonal sampling, we show the following results by induction:

$$F(X_{k_0+k}) - F(X^*) \le \left(1 - \frac{\mu}{4L}\frac{r(r-1)}{n(n-1)}\right)^{\sum_{i=0}^{k-1}Y_i}\left(F(X_{k_0}) - F(X^*)\right)$$

where $Y_i \in \{0,1\}$ is the same random variable defined above. It is clear at $k = 1$, by (10) and by the same argument as above, we have

$$F(X_{k_0+1}) - F(X^*) = F(X_{k_0+1}) - F(X_{k_0}) + F(X_{k_0}) - F(X^*)$$

$$\le -\frac{1}{8L}\frac{r(r-1)}{n(n-1)}\|\text{grad}F(X_{k_0})\|^2 Y_0 + F(X_{k_0}) - F(X^*)$$

$$\le \left(1 - \frac{\mu}{4L}\frac{r(r-1)}{n(n-1)}\right)^{Y_0}\left(F(X_{k_0}) - F(X^*)\right)$$

Now there exists an iteration $k' \geq 1$ such that for all $k < k'$, we have

$$F(X_{k_0+k}) - F(X^*) \leq \left(1 - \frac{\mu}{4L}\frac{r(r-1)}{n(n-1)}\right)^{\sum_{i=0}^{k-1} Y_i} \left(F(X_{k_0}) - F(X^*)\right).$$

Then we verify for $k = k'$,

$$
\begin{aligned}
F(X_{k_0+k'}) - F(X^*) &= F(X_{k_0+k'}) - F(X_{k_0+k'-1}) + F(X_{k_0+k'-1}) - F(X^*) \\
&\leq -\frac{1}{8L}\frac{r(r-1)}{n(n-1)}\|\mathrm{grad}F(X_{k_0+k'-1})\|^2 Y_{k'-1} + F(X_{k_0+k'-1}) - F(X^*) \\
&\leq \left(1 - \frac{\mu}{4L}\frac{r(r-1)}{n(n-1)}\right)^{Y_{k'-1}} \left(F(X_{k_0+k'-1}) - F(X^*)\right) \\
&\leq \left(1 - \frac{\mu}{4L}\frac{r(r-1)}{n(n-1)}\right)^{Y_{k'-1}} \left(1 - \frac{\mu}{4L}\frac{r(r-1)}{n(n-1)}\right)^{\sum_{i=0}^{k'-2} Y_i} \left(F(X_{k_0}) - F(X^*)\right) \\
&\leq \left(1 - \frac{\mu}{4L}\frac{r(r-1)}{n(n-1)}\right)^{\sum_{i=0}^{k'-1} Y_i} \left(F(X_{k_0}) - F(X^*)\right)
\end{aligned}
$$

where the second last inequality is by induction. This completes the induction. Then using a similar argument, we have

$$
\begin{aligned}
F(X_{k_0+k}) - F(X^*) &\leq \left(1 - \frac{\mu}{4L}\frac{r(r-1)}{n(n-1)}\right)^{\sum_{i=0}^{k-1} Y_i} \left(F(X_{k_0}) - F(X^*)\right) \\
&\leq \left(1 - \frac{\mu}{4L}\frac{r(r-1)}{n(n-1)}\right)^{(1-\delta)(1-\tau(n,r))k} \left(F(X_{k_0}) - F(X^*)\right) \\
&\leq \exp\left(-\frac{\mu}{4L}\frac{r(r-1)}{n(n-1)}(1-\delta)(1-\tau(n,r))k\right) \left(F(X_{k_0}) - F(X^*)\right)
\end{aligned}
$$

with probability at least $1 - \exp\left(-\frac{\delta^2}{2}(1-\tau(n,r))k\right)$ by (11). The proof for the permutation case is the same and thus omitted.

For simplicity, we fix $\delta = 1/2$ such that the results hold with probability at least $1 - \exp\left(-(1-\tau(n,r))k/8\right)$ for orthogonal sampling and results hold with probability at least $1 - \exp(-\binom{n}{r}^{-1}k/8)$ for permutation sampling. $\qquad\square$

### D.5. Proof of Theorem 5.7

*Proof of Theorem 5.7.* By Lemma 5.2, we see $\widetilde{F}_X(Y)$ is $L$-smooth with $L = C_0 + C_1$. Then we have for any $Y$ and $W \in T_Y \mathcal{O}(r,r)$,

$$\widetilde{F}_X(\mathrm{Retr}_Y(W)) \leq \widetilde{F}_X(Y) + \langle \mathrm{grad}\widetilde{F}_X(Y), W \rangle + \frac{L}{2}\|W\|^2.$$

Applying this inequality to the update and recalling $\widetilde{F}_k(Y) = F(U(Y)X_k)$, we have

$$
\begin{aligned}
F(X_{k+1}) = \widetilde{F}_k(\mathrm{Retr}_{I_r}(-\eta\,\mathrm{grad}\widetilde{F}(I_r))) &\leq \widetilde{F}_k(I_r) - \left(\eta - \frac{\eta^2 L}{2}\right)\|\mathrm{grad}\widetilde{F}_k(I_r)\|^2 \\
&= F(X_k) - \frac{1}{2L}\|\mathrm{grad}\widetilde{F}_k(I_r)\|^2
\end{aligned}
$$

where we note that $\widetilde{F}_k(I_r) = F(X_k)$. Taking expectation on both sides with respect to the randomness in the current iteration, we have

$$
\begin{aligned}
\mathbb{E}_k F(X_{k+1}) &\leq F(X_k) - \frac{1}{2L}\mathbb{E}_k\|\mathrm{grad}\widetilde{F}_k(I_r)\|^2 \\
&= F(X_k) - \frac{1}{2L}\frac{r(r-1)}{n(n-1)}\|\mathrm{grad}F_k(I_n)\|^2
\end{aligned}
$$

where recall $F_k(U) = F(UX_k)$.

Hence by Lemma 5.3, we have $\|\mathrm{grad}F_k(I_n)\|^2 \geq \frac{1}{2}\|\mathrm{grad}F(X_k)\|^2$ and taking full expectation,

$$\mathbb{E}[F(X_{k+1}) - F(X_k)] \leq -\frac{1}{4L}\frac{r(r-1)}{n(n-1)}\mathbb{E}[\|\mathrm{grad}F(X_k)\|^2], \tag{12}$$

Hence telescoping the inequality from $i = 0, ..., k-1$ gives

$$\frac{1}{k}\sum_{i=0}^{k-1}\mathbb{E}[\|\mathrm{grad}F(X_i)\|^2] \leq \frac{4L}{k}\frac{n(n-1)}{r(r-1)}(F(X_0) - F^*).$$

Then we notice that $\min_{i=0,...,k-1}\mathbb{E}[\|\mathrm{grad}F(X_i)\|^2] \leq \frac{1}{k}\sum_{i=0}^{k-1}\mathbb{E}[\|\mathrm{grad}F(X_i)\|^2]$ finishes the proof for the non-convex case.

To show the second result on convergence under PL condition, we notice that by Definition 5.4, and that $\min_{X\in\mathcal{U}} F(X) = F(X^*)$. Then once $X_{k_0} \in \mathcal{U}$, we can follow the same proof that $X_k \in \mathcal{U}$ for all $k \geq k_0$. Then we have using, (12) and Definition 5.4 that

$$\mathbb{E}[F(X_{k+1}) - F(X^*)] = \mathbb{E}[F(X_{k+1}) - F(X_k)] + \mathbb{E}[F(X_k) - F(X^*)]$$
$$\leq -\frac{1}{4L}\frac{r(r-1)}{n(n-1)}\mathbb{E}[\|\mathrm{grad}F(X_k)\|^2] + \mathbb{E}[F(X_k) - F(X^*)]$$
$$\leq -\frac{2\mu}{4L}\frac{r(r-1)}{n(n-1)}\mathbb{E}[F(X_k) - F(X^*)] + \mathbb{E}[F(X_k) - F(X^*)]$$
$$\leq \left(1 - \frac{\mu}{2L}\frac{r(r-1)}{n(n-1)}\right)\mathbb{E}[F(X_k) - F(X^*)].$$

Let $k_0$ be a sufficiently large iteration such that $X_{k_0} \in \mathcal{U}$. Then, we have $\mathbb{E}[F(X_{k_0+k}) - F(X^*)] \leq \exp(-\frac{\mu}{2L}\frac{r(r-1)}{n(n-1)}k)\mathbb{E}[F(X_k) - F(X^*)]$, where we use $(1-a)^k \leq \exp(-ak)$ for $k > 0$. $\qquad\square$

## D.6. Proof of Theorem 5.13

*Proof of Theorem 5.13.* From Lemma 5.2, we know that $F$ is $L$-smooth, where $L = C_0 + C_1$. Then

$$F(X_{k+1}) = \widetilde{F}_k(\mathrm{Retr}_{I_r}(-\eta\mathrm{grad}\tilde{f}_k(I_r;\xi_k)))$$
$$\leq \widetilde{F}_k(I_r) - \eta\langle\mathrm{grad}\widetilde{F}_k(I_r), \mathrm{grad}\tilde{f}_k(I_r;\xi_k)\rangle + \frac{\eta^2 L}{2}\|\mathrm{grad}\tilde{f}_k(I_r;\xi_k)\|^2.$$

Taking expectation with respect to $\xi_k$, we obtain

$$\mathbb{E}_{\xi_k}[F(X_{k+1})] \leq \mathbb{E}_{\xi_k}[F(X_k)] - \eta\|\mathrm{grad}\widetilde{F}_k(I_r)\|^2 + \frac{\eta^2 L}{2}\mathbb{E}_{\xi_k}\|\mathrm{grad}\tilde{f}_k(I_r;\xi_k)\|^2,$$

where we notice $\widetilde{F}_k(I_r) = F(X_k)$ and use the unbiasedness assumption. In addition, we can bound

$$\mathbb{E}_{\xi_k}\|\mathrm{grad}\tilde{f}_k(I_r;\xi_k) - \mathrm{grad}\widetilde{F}_k(I_r)\|^2$$
$$= \mathbb{E}_{\xi_k}\|P_k(r)\big(\mathrm{grad}f_k(I_n;\xi_k) - \mathrm{grad}F_k(I_n)\big)P_k(r)^\top\|^2$$
$$\leq \frac{1}{4}\mathbb{E}_{\xi_k}\|\big(\nabla f(X_k;\xi_k)X_k^\top - \nabla F(X_k)X_k^\top\big) + \big(X_k\nabla F(X_k)^\top - X_k\nabla f(X_k;\xi_k)\big)\|^2$$
$$\leq \mathbb{E}_{\xi_k}\|\nabla f(X_k;\xi_k) - \nabla F(X_k)\|^2 \leq \sigma^2$$

where we use the definition of $\tilde{f}_k, f_k, \widetilde{F}_k, F_k$ and orthogonality of $P_k(r)$ and $X_k$. The last inequality is by bounded variance assumption.

Then we further expand

$$\mathbb{E}_{\xi_k}\|\mathrm{grad}\tilde{f}_k(I_r;\xi_k)\|^2 = \mathbb{E}_{\xi_k}\|\mathrm{grad}\widetilde{F}_k(I_r)\|^2 + \mathbb{E}_{\xi_k}\|\mathrm{grad}\tilde{f}_k(I_r;\xi_k) - \mathrm{grad}\widetilde{F}_k(I_r)\|^2$$
$$\leq \|\mathrm{grad}\widetilde{F}_k(I_r)\|^2 + \sigma^2,$$

where we use the unbiasedness in the first equality. This gives

$$\mathbb{E}_{\xi_k}[F(X_{k+1})] \leq \mathbb{E}_{\xi_k}[F(X_k)] - \left(\eta - \frac{\eta^2 L}{2}\right)\|\mathrm{grad}\widetilde{F}_k(I_r)\|^2 + \frac{\eta^2 L\sigma^2}{2}.$$

Now taking expectation with respect to the randomness in $P_k$, we can show from Proposition 5.5 that

$$\mathbb{E}_k[F(X_{k+1})] \leq F(X_k) - \left(\eta - \frac{\eta^2 L}{2}\right)\frac{r(r-1)}{n(n-1)}\|\mathrm{grad}F_k(I_n)\|^2 + \frac{\eta^2 L\sigma^2}{2}$$

$$\leq F(X_k) - \left(\eta - \frac{\eta^2 L}{2}\right)\frac{r(r-1)}{2n(n-1)}\|\mathrm{grad}F(X_k)\|^2 + \frac{\eta^2 L\sigma^2}{2},$$

where we denote $\mathbb{E}_k$ to represent the expectation over randomness in iteration $k$ and the second inequality is by Lemma 5.3. Taking full expectation, we obtain

$$\mathbb{E}[F(X_{k+1}) - F(X_k)] \leq -\left(\eta - \frac{\eta^2 L}{2}\right)\frac{r(r-1)}{2n(n-1)}\mathbb{E}\|\mathrm{grad}F(X_k)\|^2 + \frac{\eta^2 L\sigma^2}{2} \tag{13}$$

Rearranging the terms and summing over $k \in [K]$ gives

$$\left(\eta - \frac{\eta^2 L}{2}\right)\frac{r(r-1)}{2n(n-1)} \sum_{k=0}^{K-1} \mathbb{E}\|\mathrm{grad}F(X_k)\|^2 \leq F(X_0) - F^* + \frac{K\eta^2 L\sigma^2}{2}.$$

Choosing $\eta = \min\{L^{-1}, c\sigma^{-1}K^{-1/2}\}$ for some constant $C > 0$. Then $\eta - \eta^2 L/2 \geq \eta/2$ and thus we can show

$$\frac{1}{K}\sum_{k=0}^{K-1} \mathbb{E}\|\mathrm{grad}F(X_k)\|^2 \leq \frac{1}{K}\frac{4n(n-1)}{r(r-1)}\max\{L, \sigma\sqrt{K}/c\}\Delta_0 + \frac{4n(n-1)}{r(r-1)}\eta L\sigma^2$$

$$\leq \frac{4n(n-1)}{r(r-1)}\left(\frac{L\Delta_0}{K} + \frac{\sigma\Delta_0}{c\sqrt{K}} + \frac{L\sigma c}{\sqrt{K}}\right)$$

$$= \frac{4n(n-1)}{r(r-1)}\left(\frac{L\Delta_0}{K} + \frac{2\sigma\sqrt{\Delta_0 L}}{\sqrt{K}}\right)$$

where we let $\Delta_0 = F(X_0) - F^*$ and we choose $c = \sqrt{\Delta_0/L}$ to minimize the upper bound. Finally, noticing $\min_{i=0,\ldots K-1} \mathbb{E}\|\mathrm{grad}F(X_k)\|^2 \leq \frac{1}{K}\sum_{k=0}^{K-1} \mathbb{E}\|\mathrm{grad}F(X_k)\|^2$ completes the proof under nonconvex loss.

To show the second result on convergence under PL condition, suppose $X_k \in \mathcal{U}$. For such $k$, we have by (13)

$$\mathbb{E}[F(X_{k+1}) - F(X^*)] = \mathbb{E}[F(X_{k+1}) - F(X_k)] + \mathbb{E}[F(X_k) - F(X^*)]$$

$$\leq -\left(\eta - \frac{\eta^2 L}{2}\right)\frac{r(r-1)}{2n(n-1)}\mathbb{E}\|\mathrm{grad}F(X_k)\|^2 + \frac{\eta^2 L\sigma^2}{2} + \mathbb{E}[F(X_k) - F(X^*)]$$

$$\leq -\left(\eta - \frac{\eta^2 L}{2}\right)\frac{r(r-1)}{n(n-1)}\mu\mathbb{E}[F(X_k) - F(X^*)] + \frac{\eta^2 L\sigma^2}{2} + \mathbb{E}[F(X_k) - F(X^*)]$$

$$= \left(1 - \mu(\eta - \eta^2 L/2)\frac{r(r-1)}{n(n-1)}\right)\mathbb{E}[F(X_k) - F(X^*)] + \frac{\eta^2 L\sigma^2}{2}$$

Given $\eta \leq L^{-1}$, we obtain $\mathbb{E}[F(X_{k+1}) - F(X^*)] \leq \left(1 - \frac{\mu}{2L}\frac{r(r-1)}{n(n-1)}\right)\mathbb{E}[F(X_k) - F(X^*)] + \frac{\sigma^2}{2L}$. Then for $k_0, \ldots, k_1$ such that $X_{k_0}, \ldots, X_{k_1} \in \mathcal{U}$, we have

$$\mathbb{E}[F(X_{k_1}) - F(X^*)]$$

$$\leq \left(1 - \frac{\mu}{2L}\frac{r(r-1)}{n(n-1)}\right)^{k_1-k_0}\mathbb{E}[F(X_{k_0}) - F(X^*)] + \frac{\sigma^2}{2L}\sum_{i=0}^{k_1-k_0-1}\left(1 - \frac{\mu}{2L}\frac{r(r-1)}{n(n-1)}\right)^i$$

$$\leq \left(1 - \frac{\mu}{2L}\frac{r(r-1)}{n(n-1)}\right)^{k_1-k_0}\mathbb{E}[F(X_{k_0}) - F(X^*)] + \frac{\sigma^2}{\mu}\frac{n(n-1)}{r(r-1)}$$

$$\leq \exp\left(-\frac{\mu}{2L}\frac{r(r-1)}{n(n-1)}(k_1 - k_0)\right)\mathbb{E}[F(X_{k_0}) - F(X^*)] + \frac{\sigma^2}{\mu}\frac{n(n-1)}{r(r-1)}$$

where the second inequality is by $\sum_{i=0}^{k-1}(1-\alpha)^i \leq \frac{1}{\alpha}$. □

# E. Extension to finite-sum settings

In this section, we generalize the analysis of Section 5.3 to mini-batch settings, namely

$$\min_{X \in \mathrm{St}(n,p)} \left\{ F(X) = \frac{1}{N} \sum_{i=1}^{N} f_i(X) \right\}, \tag{14}$$

where $f_i, i \in [N]$ represents $N$ component functions.

We accordingly modify Algorithm 1 by sampling $B_k \subset [N]$ uniformly from $[N]$ with replacement. Without loss of generality, we set $|B_k| = B$ for all $k$. Then we replace the Riemannian mini-batch gradient as

$$\mathrm{grad}\widetilde{f}_{B_k}(I_r) := \frac{1}{B} \sum_{i \in B_k} \mathrm{grad}\widetilde{f}_i^k(I_r),$$

where we use $\widetilde{f}_i^k(Y) = f_i(U_k(Y)X_k)$ similarly as before. Then we update the iterate with the mini-batch gradient $\mathrm{grad}\widetilde{f}_{B_k}(I_r)$. Notice that by the definition, we have $\widetilde{F}_k(Y) = \frac{1}{N}\sum_{i=1}^{N}\widetilde{f}_i^k(Y)$. Because $B_k$ is sampled uniformly with replacement, we can easily see that $\mathbb{E}_{B_k}[\mathrm{grad}\widetilde{f}_{B_k}(I_r)] = \mathrm{grad}\widetilde{F}_k(I_r)$.

Before we analyze the convergence, we require the following assumption on bounded variance.

**Assumption E.1.** The gradient of each component function has bounded variance, i.e., $\mathbb{E}\|\nabla f_i(X) - \nabla F(X)\|^2 \leq \sigma^2$ for any $i \in [N]$ and $X \in \mathrm{St}(n,p)$.

**Theorem E.2.** *Consider mini-batch setting as in (14) and solve with mini-batch gradient descent with batch size $B$. Under Assumption 5.1 and E.1, suppose we choose $\eta = \min\{L^{-1}, \sqrt{B\Delta_0/L}\sigma^{-1}K^{-1/2}\}$, where we denote $\Delta_0 = F(X_0) - F^*$. Then we can show*

$$\min_{i=0,\dots K-1} \mathbb{E}\|\mathrm{grad}F(X_k)\|^2 \leq \frac{4n(n-1)}{r(r-1)} \left( \frac{L\Delta_0}{K} + \frac{2\sigma\sqrt{\Delta_0 L}}{\sqrt{KB}} \right).$$

*Suppose there exist $X_{k_0}, \dots, X_{k_1} \in \mathcal{U}$ for some $k_1 > k_0$, where $\mathcal{U}$ is defined in Theorem 5.7. Then we have $\mathbb{E}[F(X_{k_1}) - F(X^*)] \leq \exp\left(-\frac{\mu}{2L}\frac{r(r-1)}{n(n-1)}(k_1 - k_0)\right)\mathbb{E}[F(X_{k_0}) - F(X^*)] + \frac{\sigma^2}{B\mu}\frac{n(n-1)}{r(r-1)}$.*

*Proof of Theorem E.2.* From Lemma 5.2, we know that $F$ is $L$-smooth, where $L = C_0 + C_1$. Then

$$F(X_{k+1}) = \widetilde{F}_k(\mathrm{Retr}_{I_r}(-\eta\mathrm{grad}\widetilde{f}_{B_k}(I_r)))$$

$$\leq \widetilde{F}_k(I_r) - \eta\langle\mathrm{grad}\widetilde{F}_k(I_r), \mathrm{grad}\widetilde{f}_{B_k}(I_r)\rangle + \frac{\eta^2 L}{2}\|\mathrm{grad}\widetilde{f}_{B_k}(I_r)\|^2.$$

Taking expectation with respect to $B_k$, we obtain

$$\mathbb{E}_{B_k}[F(X_{k+1})] \leq \mathbb{E}_{B_k}[F(X_k)] - \eta\|\mathrm{grad}\widetilde{F}_k(I_r)\|^2 + \frac{\eta^2 L}{2}\mathbb{E}_{B_k}\|\mathrm{grad}\widetilde{f}_{B_k}(I_r)\|^2,$$

where we notice $\widetilde{F}_k(I_r) = F(X_k)$ and by the unbiasedness. In addition, we can bound

$$\mathbb{E}_{B_k}\|\mathrm{grad}\widetilde{f}_{B_k}(I_r) - \mathrm{grad}\widetilde{F}_k(I_r)\|^2$$

$$= \mathbb{E}_{B_k}\|P_k(r)\big(\mathrm{grad}f_{B_k}(I_n) - \mathrm{grad}F_k(I_n)\big)P_k(r)^\top\|^2$$

$$\leq \frac{1}{4}\mathbb{E}_{B_k}\|\big(\nabla f_{B_k}(X_k)X_k^\top - \nabla F(X_k)X_k^\top\big) + \big(X_k\nabla F(X_k)^\top - X_k\nabla f_{B_k}(X_k)\big)\|^2$$

$$\leq \mathbb{E}_{B_k}\|\nabla f_{B_k}(X_k) - \nabla F(X_k)\|^2$$

$$= \frac{1}{B^2}\sum_{i \in B_k}\mathbb{E}\|\nabla f_i(X_k) - \nabla F(X_k)\|^2 \leq \frac{\sigma^2}{B},$$

where the second last inequality is by independence of samples in $B_k$ and the last inequality is by bounded variance assumption (Assumption E.1).

The subsequence analysis follows exactly the same as in Theorem 5.13, where we replace $\sigma^2$ with $\sigma^2/B$. □

# F. On the exact convergence of RSDM

In this section, we examine the exact convergence of RSDM. We require the projection matrices for the inner iterations satisfy the condition (4), which we recall below:

$$\sum_{s=0}^{S-1} \|P_k^s(r)\text{grad}F_k(I_n)P_k^s(r)^\top\|^2 \geq C_p\|\text{grad}F_k(I_n)\|^2$$

for some constant $C_p > 0$. This is a non-degenerate condition over the selection of random matrix $P_k^s$ over certain iterations such that projected gradient does not vanish.

Let us define by $\mathcal{S}_n^r$ the set of all truncated permutation matrix $P(r) \in \mathbb{R}^{r \times n}$. To each element $P$ of $\mathcal{S}_n^r$, we can associate a unique truncated permutation $\pi$ defined by

$$\forall i \leq r : \pi(i) \text{ is such that } P^\top e_i = e_{\pi(i)}.$$

Notice that $\pi$ is defined only for the first $r$ integers as the matrix $P \in \mathcal{S}_n^r$ has only $r$ rows.

**Proposition F.1.** *Let* $S = \frac{n!}{(n-r)!}$, *and assume that the matrices* $\{P_k^s\}_{s=0}^{S-1}$ *are randomly sampled, without replacement, from* $\mathcal{S}_n^r$ *(at each iteration $s$, we pick randomly a matrix from $\mathcal{S}_n^r$ that has not already been chosen) then condition (4) holds with* $C_p = \frac{(n-2)!r(r-1)}{(n-r)!}$.

*Proof of Proposition F.1.* The proof follows the same idea with the proof of Proposition 5.5 in the permutation case. Indeed let us fix a matrix $P_k^s(r)$ in $\mathcal{S}_n^r$, with associated permutation $\pi$. Using the fact that $\text{grad}F_k(I_n)$ is skew symmetric, we have

$$\|P_k^s(r)\text{grad}F_k(I_n)P_k^s(r)^\top\|^2 = 2 \sum_{1 \leq i < j \leq r} (\text{grad}F_k(I_n))_{(\pi(i),\pi(j))}^2 = \sum_{1 \leq i,j \leq r} (\text{grad}F_k(I_n))_{(\pi(i),\pi(j))}^2.$$

Hence, we can write that

$$\sum_{s=0}^{S-1} \|P_k^s(r)\text{grad}F_k(I_n)P_k^s(r)^\top\|^2 = \sum_{\pi \in \mathcal{S}_n^r} \sum_{1 \leq i,j \leq r} (\text{grad}F_k(I_n))_{(\pi(i),\pi(j))}^2.$$

Notice that we obtain the same expression as in the proof of Proposition 5.5 but without the factor $\frac{n!}{(n-r)!}$. Indeed, the summation on $s$ term in the above equation corresponds to what we denoted by $\mathbb{E}\|\text{grad}\widetilde{F}_k(I_r)\|^2$ in the proof of the proposition. Hence following the same argument, we have that

$$\sum_{s=0}^{S-1} \|P_k^s(r)\text{grad}F_k(I_n)P_k^s(r)^\top\|^2 = \frac{n!}{(n-r)!}\frac{r(r-1)}{n(n-1)}\|\text{grad}F_k(I_n)\|^2,$$

that is

$$\sum_{s=0}^{S-1} \|P_k^s(r)\text{grad}F_k(I_n)P_k^s(r)^\top\|^2 = \frac{(n-2)!r(r-1)}{(n-r)!}\|\text{grad}F_k(I_n)\|^2.$$

$\square$

Next, we analyze the exact convergence if we sample according to condition (4).

Before we derive the results, we recall the notation that $\widetilde{F}_k^s(Y) := F(U_k^s(Y)X_k^s)$ and $F_k^s(U) := F(UX_k^s)$. We also require the following lemma from (Chen et al., 2020) that bounds the retraction on Stiefel manifold with the Euclidean retraction, i.e., addition.

**Lemma F.2** (Chen et al. (2020))**.** *For all* $X \in \text{St}(n,p)$ *and* $U \in T_X\text{St}(n,p)$, *there exists a constant $M > 0$ such that* $\|\text{Retr}_X(U) - X\| \leq M\|U\|$.

*Proof of Theorem 5.11.* Following the analysis of Theorem 5.7, we have for any $k$,

$$F(X_k^{s+1}) \le F(X_k^s) - \frac{1}{2L}\|\mathrm{grad}\widetilde{F}_k^s(I_r)\|^2. \tag{15}$$

Next, recall that $\mathrm{grad}F_k^s(I_n) = (\nabla F(X_k^s) - \nabla F(X_k^s)^\top)/2$. Then we show for any $s = 0, ..., S-1$ and any $k$,

$$\|\mathrm{grad}F_k^s(I_n) - \mathrm{grad}F_k(I_n)\| \le \|\nabla F(X_k^s) - \nabla F(X_k)\| \le C_1\|X_k^s - X_k\|$$

$$\le C_1\sum_{i=0}^{s-1}\|X_k^{i+1} - X_k^i\|$$

$$= C_1\sum_{i=0}^{s-1}\|P_k^i(r)(Y_k^i - I_r)P_k^i(r)X_k^i\|$$

$$\le C_1\sum_{i=0}^{s-1}\|Y_k^i - I_r\|$$

$$\le C_1\eta M\sum_{i=0}^{s-1}\|\mathrm{grad}\widetilde{F}_k^i(I_r)\| \tag{16}$$

where the first and third inequalities are by triangle inequality and the second inequality is by Assumption 5.1 that (Euclidean) Hessian is upper bounded. The fourth inequality is by the orthogonality of $P_k^i(r)$ and $X_k^i$. The last inequality is by Lemma F.2.

Then we can bound for any $k, s$

$$\|P_k^s(r)\mathrm{grad}F_k(I_n)P_k^s(r)^\top\|^2$$
$$\le 2\|P_k^s(r)\mathrm{grad}F_k^s(I_n)P_k^s(r)^\top\|^2 + 2\|P_k^s(r)(\mathrm{grad}F_k^s(I_n) - \mathrm{grad}F_k(I_n))P_k^s(r)^\top\|^2$$
$$\le 2\|P_k^s(r)\mathrm{grad}F_k^s(I_n)P_k^s(r)^\top\|^2 + 2C_1^2\eta^2 M^2 S\sum_{i=0}^{s-1}\|\mathrm{grad}\widetilde{F}_k^i(I_r)\|^2.$$

where the second inequality is by (16). Summing over the above inequality yields

$$\sum_{s=0}^{S-1}\|P_k^s(r)\mathrm{grad}F_k(I_n)P_k^s(r)^\top\|^2 \le (2 + 2C_1^2\eta^2 M^2 S^2)\sum_{s=0}^{S-1}\|P_k^s(r)\mathrm{grad}F_k^s(I_n)P_k^s(r)^\top\|^2$$

$$= (2 + 2C_1^2\eta^2 M^2 S^2)\sum_{s=0}^{S-1}\|\mathrm{grad}\widetilde{F}_k^s(I_r)\|^2. \tag{17}$$

Finally, we sum over the inequality (15) for $s = 0, ..., S-1$, which obtains

$$F(X_k^S) = F(X_k^0) - \frac{1}{2L}\sum_{s=0}^{S-1}\|\mathrm{grad}\widetilde{F}_k^s(I_r)\|^2$$

$$\le F(X_k^0) - (L^{-1} + C_1^2 L^{-3}M^2 S^2)\sum_{s=0}^{S-1}\|P_k^s(r)\mathrm{grad}F_k(I_n)P_k^s(r)^\top\|^2$$

$$\le F(X_k^0) - C_p(L^{-1} + C_1^2 L^{-3}M^2 S^2)\|\mathrm{grad}F_k(I_n)\|^2$$

$$\le F(X_k^0) - C_p(L^{-1} + C_1^2 L^{-3}M^2 S^2)/2\|\mathrm{grad}F(X_k)\|^2$$

where the second inequality is by (17) and $\eta = 1/L$. The second inequality is by (4). The last inequality is by Lemma 5.3. Noticing tht $X_k^S = X_{k+1}$ and $X_k^0 = X_k$ and telescoping the result, we have

$$\frac{1}{k}\sum_{i=0}^{k-1}\|\mathrm{grad}F(X_i)\|^2 \le \frac{1}{k}\frac{2L}{C_p(1 + C_1^2 L^{-2}M^2 S^2)}(F(X_0) - F^*),$$

which shows the desired result. $\qquad\square$

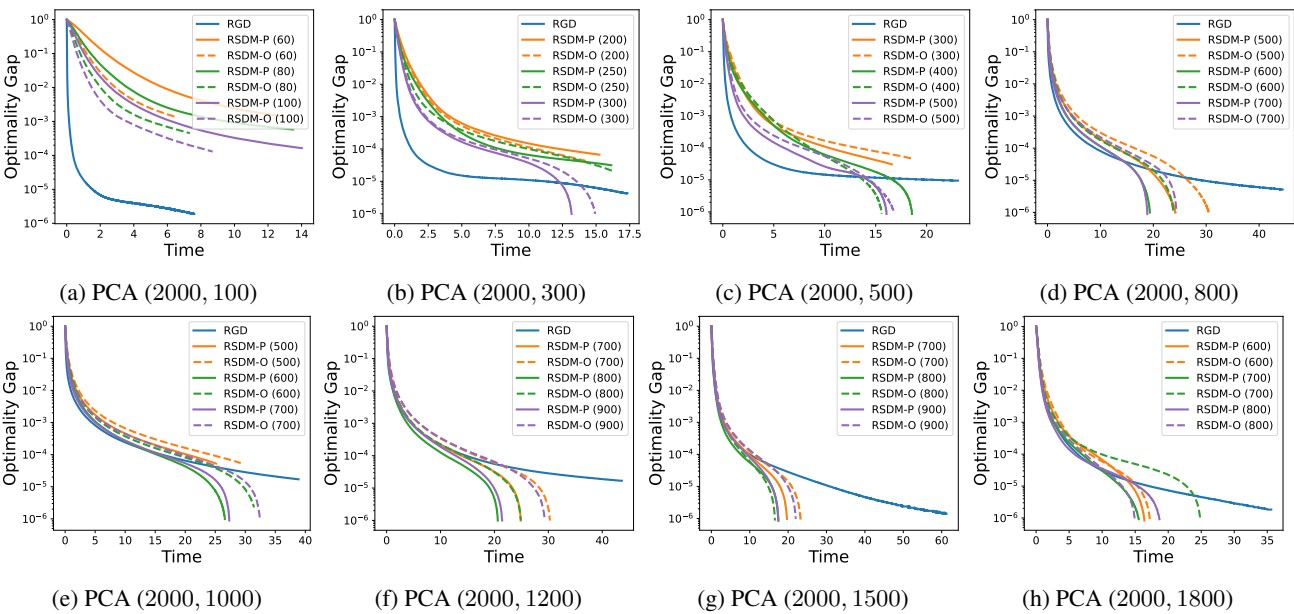

Figure 8: Experiments on PCA problem with various settings of $p$. The numbers in brackets correspond to $n, p$ respectively.

## G. Additional numerical experiments on different $p$

In this section, we investigate the performance of proposed RSDM across various settings of $p$ on the PCA problem. Following Section 6.2, we employ the same procedures in generating the data and fix dimension $n = 2000$. We sweep across a range of values of $p$, i.e., $p = 100, 300, 500, 800, 1000, 1200, 1500, 1800$. For each problem instance, we also run RSDM with different submanifold size $r$.

The convergence of RSDM in comparison to RGD is given in Figure 8. We observe that indeed when $p$ becomes small, the performance gap between RSDM and RGD is decreasing, which is in accordance with derived total complexities. Nevertheless, we still observe that RSDM is able to outperform RGD across all the settings, except for the case when $p = 100$. This validates the benefits of RSDM even when $p$ becomes smaller. Nonetheless, we admit that when $p$ becomes significantly small relative to $n$ (as in the case when $p = 100$), RGD may perform better because the cost of retraction is much less pronounced. However, this motivates a hybrid design of RSDM such that when $p$ is relatively small, it effectively behaves similarly to RGD. This is left for future exploration.

An interesting observation is that when $p = 300$ and $p = 500$, selecting $r = p$ can still yield significantly improved convergence especially near optimality. We conjecture this is due to the randomized submanifold descent leads to better-conditioned loss landscape around optimality, and thus performs well particularly for ill-conditioned problems. The theoretical analysis of such claim is left for future works.

## H. Comparison to RSSM

In this section, we compare our method with RSSM (Cheung et al., 2024), which can be viewed as projected Euclidean coordinate descent on the Stiefel manifold. In particular, Algorithm 1 of (Cheung et al., 2024) translates into the following update steps for smooth optimization. For each iteration $k$,

1. Pick index set $C \subset [p]$ with no repetition in $C$.

2. Compute partial Euclidean gradient and project to tangent space:

$$\mathrm{grad}_C F(X^k) = X_C \mathrm{skew}(X_C^\top \nabla_C F(X^k)) + (I - XX^\top)\nabla_C F(X^k),$$

where $X_C \in \mathbb{R}^{n \times |C|}$ is the columns of $X$ corresponding to the index in $C$ and $\mathrm{skew}(A) = (A - A^\top)/2$ denotes the skew-symmetric operation. $\nabla_C F(X)$ is the partial Euclidean gradient with respect to the columns of $X$ in $C$.

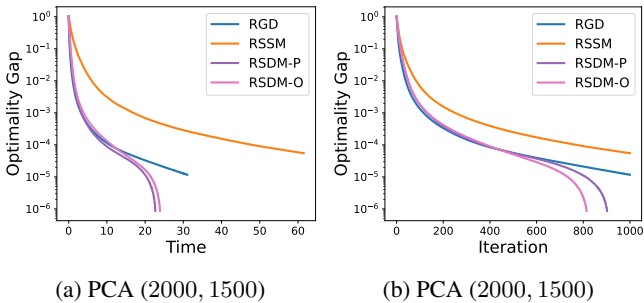

(a) PCA (2000, 1500)  (b) PCA (2000, 1500)

Figure 9: Comparison of proposed RSDM with RSSM (Cheung et al., 2024) on the PCA problem. We observe RSDM converges significantly faster than RSSM.

3. Update the columns of $X_k$ in $C$ by projected gradient descent while keeping other columns the same:

$$X_C^{k+1} = \text{Proj}_{\text{St}(n,|C|)}(X_C^k - \eta \text{grad}_C F(X^k)),$$

where $\text{Proj}_{\text{St}(n,p)}(\cdot)$ denotes the projection from Euclidean space to Stiefel manifold via SVD.

**Comparison to proposed RSDM.** RSSM can be viewed as Euclidean coordinate descent projected to Stiefel manifold. Let $r = |C| < p$ be the number of columns sampled. The gradient computation requires $O(npr)$ complexity and iterate update requires $O(nr^2)$ for non-standard linear algebra operation. Thus the per-iteration complexity is costly than our proposed RSDM (with permutation sampling), which requires $O(nr^2)$ for gradient complexity and $O(r^3)$ for non-standard linear algebra operation. Thus, we see RSDM-P requires much per-iteration complexity compared to RSSM. In addition, apart from the advantages in per-iteration complexity, RSDM also allows easy generalization to quotient manifolds, such as Grassmann manifold, while this appears challenging for RSSM due to its operations along the columns.

**Numerical comparisons.** To further validate the benefits of RSDM to RSSM (Cheung et al., 2024), we have implemented RSSM with a fixed stepsize and choose $C$ from $[p]$ uniformly without repetition.[1] We have tuned both $r = |C|$ and stepsize $\eta$ for RSSM to the best performance. We compare the performance on the PCA problem where we tune $r = 700$ and $\eta = 0.1$ for RSSM.

The results are included in Figure 9. We notice that RSDM (either with orthogonal or permutation sampling) achieves significantly faster convergence compared to RSSM. This verifies the numerical benefits of RSDM over RSSM.

## I. Comparison to OBCD

This section compares proposed RSDM to OBCD (Yuan, 2023). We first remark that (Yuan, 2023) is primarily designed for nonsmooth optimization and thus optimality conditions and convergence analysis are largely different. Here we adapt the algorithm of OBCD to the smooth case.

In this case, because they parameterize $X_{k+1} = X_k + U_B(V - I_r)U_B^\top X_k$ where $U_B \in \mathbb{R}^{n \times r}$ is a random truncated permutation matrix, and $V \in \mathbb{R}^{r \times r}$, they minimize a quadratic upper bound for $F(X_k + U_B(V - I_r)U_B^\top X_k)$. Suppose $F$ is $L_F$ smooth, then the subproblem translates into

$$\min_{V \in \text{St}(r,r)} \langle V - I_r, U_B^\top \nabla F(X_k) X_k^\top U_B \rangle + \frac{L_F}{2} \|V - I_r\|^2 \tag{18}$$

for some constant $L_F$ that depends on the smoothness of $F$. And thus there exists a global solution to (18), i.e., $V^*$ is the top $r$ eigenvectors of $I_r - \frac{1}{L_F} U_B^\top \nabla F(X_k) X_k^\top U_B$.

This is related but different to our update of $Y$ (according to our notation) when we use permutation sampling strategy. In particular, we update $Y$ by

$$Y = \text{Retr}_{I_r}(-\eta \text{grad} \widetilde{F}_k(I_r)) = \text{Retr}_{I_r}\left(-\frac{\eta}{2} P_k(r)(\nabla F(X_k) X_k^\top - X_k \nabla F(X_k)^\top) P_k(r)^\top\right).$$

---

[1]It is worth mentioning that (Cheung et al., 2024) did not include any numerical experiments nor provide the code.

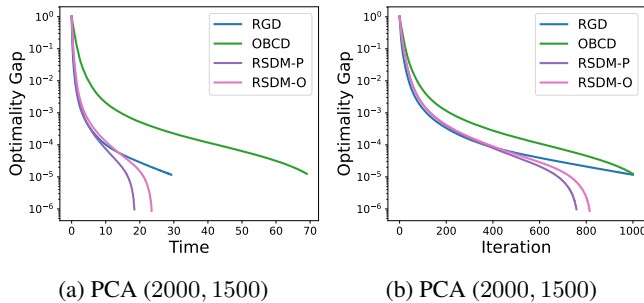

(a) PCA (2000, 1500)  (b) PCA (2000, 1500)

Figure 10: Comparison of proposed RSDM with OBCD (Yuan, 2023) on the PCA problem. We observe RSDM converges significantly faster than OBCD.

The key difference is that we have a skew-symmetric operation for $\nabla F(X_k)X_k^\top$, which renders the update direction properly defined as Riemannian gradient on $\mathcal{O}(r)$. In contrast, OBCD leverages the Euclidean gradient for the update.

This difference leads to a large deviation in the proof strategy, making the analysis of (Yuan, 2023) less aligned with common analysis on Riemannian manifolds. This makes their developments more difficult to generalize to other manifolds of interest and incorporate additional optimization techniques on manifolds, such as adaptive gradients, acceleration, Newton based methods, etc.

Apart from this main difference, we also summarize other differences of their developments compared to this work:

- They show convergence to block-$k$ stationary points, which seems to be weaker than our established convergence to stationary points (as shown in Theorem 5.5 of (Yuan, 2023).

- Their convergence rate in Theorem 6.3 depends on a large binomial coefficient $C_n^r$ while our convergence has a coefficient $n^2 r^{-2}$.

- They only consider $U_B$ to be (truncated) permutation while we consider both permutation and general orthogonal matrix.

- We have shown convergence in stochastic settings and shown extension to other quotient manifolds, which is not the case for (Yuan, 2023).

- They only show convergence in expectation while we show convergence both in expectation and with high probability and almost surely.

Finally, we compare the proposed RSDM to OBCD numerically on the PCA problem. We have solved $V$ from (18) analytically with SVD. We choose $r = 700$, which is the same as RSDM for comparability and tune stepsize accordingly. The convergence plots are given in Figure 10 where we observe that OBCD (Yuan, 2023) converges significantly slower compared to RSDM. This suggests the critical difference in the update directions (Riemannian gradient for proposed RSDM and Euclidean gradient for OBCD) has led to significant convergence disparities, thus verifying superiority of the framework of Riemannian optimization employed by RSDM in this paper.

## J. RSDM with momentum

In this section, we explore the potential of RSDM when coupled with momentum. We adopt the strategy of fixing $P_k$ for several iterations, where we apply momentum. This is equivalent to taking multiple gradient descent (with momentum for minimizing $\widetilde{F}_k(Y)$ initialized from $I_r$. The procedures are included in Algorithm 3.

---

**Algorithm 3** RSDM-momentum

---

1: Initialize $X_0 \in \mathrm{St}(n, p)$.
2: **for** $k = 0, ..., K - 1$ **do**
3:     Sample $P_k \in \mathcal{O}(n)$ and let $\widetilde{F}_k(Y) = F(U_k(Y)X_k)$ where $U_k(Y)$ is defined in (2).
4:     Set $Y_k^0 = I_r$.
5:     **for** $s = 0, ...., S - 1$ **do**
6:         Compute Riemannian gradient $\mathrm{grad}\widetilde{F}_k(Y_k^s)$.
7:         Update $Y_k^{s+1} = \mathrm{Retr}_{Y_k^s}(-\eta \, \mathrm{grad}\widetilde{F}_k(Y_k^s) + \beta \mathrm{P}_{Y_k^s}(Y_k^s - Y_k^{s-1}))$.
8:     **end for**
9:     Set $X_{k+1} = U_k(Y_k^S)X_k$.
10: **end for**

---

It is worth mentioning that we now require to compute the gradient $\mathrm{grad}\widetilde{F}_k(Y)$ for any $Y \in \mathcal{O}(r)$, while previously we only need to compute $\mathrm{grad}\widetilde{F}_k(I_r)$. Specifically, we compute

$$\mathrm{grad}\widetilde{F}_k(Y) = \frac{1}{2}\big(\nabla \widetilde{F}_k(Y) - Y\nabla \widetilde{F}_k(Y)^\top Y\big)$$
$$= \frac{1}{2}\big(P_k(r)\nabla F(Z_k)X_k^\top P_k(r)^\top - YP_k(r)X_k\nabla F(Z_k)^\top P_k(r)^\top Y\big)$$

where $Z_k = U_k(Y)X_k = X_k + P_k(r)^\top(Y - I_r)P_k(r)X_k$.

To test the feasibility of the proposed algorithm, we evaluate RSDM with momentum (RSDM-M) with permutation sampling and compare against RGD (RGD-M) with momentum on the PCA problem. We consider the setting of $n = 2000, p = 1500$ and $r = 700$. We set the momentum parameter to be $0.5$ for both RSDM and RGD. We tune and set the learning rate of $0.1$ for RGD-M and $1.0$ for RSDM-M.

From Figure 11, we see that the RSDM with momentum improves the convergence of RSDM, which demonstrate the potential of incorporating momentum into our framework.

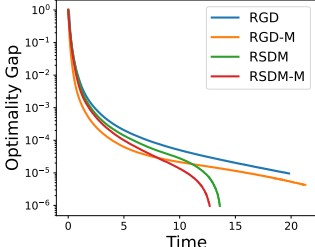

Figure 11: Comparison of RSDM with momentum with RGD with momentum on the PCA problem.

