# OpenReview forum: "Efficient Optimization with Orthogonality Constraint: a Randomized Riemannian Submanifold Method"
_ICML.cc/2025/Conference — ICML 2025 poster_

### Official Review · Reviewer_N9y9 · 2025-03-12

**Overall Recommendation:** 4

**Summary:**

In the work, the authors propose a randomized Riemannian submanifold approach for optimization on Stiefel manifolds. The authors prove that the proposed method converges under certain conditions.
Empirical results are included to support the effectiveness of the proposed method.

**Claims And Evidence:**

The claims are sound and solid.

**Essential References Not Discussed:**

N/A

**Experimental Designs Or Analyses:**

The experimental designs make sense.

**Methods And Evaluation Criteria:**

The proposed method makes sense. The authors properly evaluate their method on several problems.
The empirical results are convincing.

**Other Comments Or Suggestions:**

N/A

**Other Strengths And Weaknesses:**

This work is interesting and original.

**Questions For Authors:**

It seems that the proposed method requires an efficient computation of a Riemannian gradient on a submanifold.
The authors should comment on whether the proposed idea requires a compatible Riemannian metric to effectively reduce the computational cost.

**Relation To Broader Scientific Literature:**

Optimization

**Theoretical Claims:**

Convergence analysis is also provided to support the soundness of the proposed method.
I do not check the proofs in the appendices.

---

> ### Author Rebuttal · Authors · 2025-03-29
>
> We sincerely thank the reviewer for the positive evaluation of our work. We are glad that the reviewer found our work to be interesting and original.
>
>
> **1. Whether the proposed idea requires a compatible Riemannian metric to effectively reduce the computational cost.**
>
> We thank the reviewer for the question. In our work, we choose the Euclidean metric as the Riemannian metric for the Stiefel manifold. Under such a metric, we have derived the expression for the Riemannian gradient on the submanifold, which can be efficiently computed. We agree that using alternative metric, such as canonical metric may be interesting, which we leave for future exploration.

---

### Official Review · Reviewer_awSm · 2025-03-13

**Overall Recommendation:** 3

**Summary:**

This paper improves the efficiency of the retraction operation in the geometric optimization algorithms over the Stiefel manifold by constraining the optimization into a randomly sampled smaller manifold. Specifically, for an element X in the Stiefel manifold, it learns an orthonormal matrix U that acts on X, where only a random subspace in U is learnable. The computation complexity is reduced if the dimension of the random subspace is sufficiently small. The convergence of the proposed algorithm is analyzed, and experiments on four settings show some improvement in the convergence speed.

**Claims And Evidence:**

The claims are supported by evidence.

**Essential References Not Discussed:**

None.

**Experimental Designs Or Analyses:**

The experiment design makes sense, except for the MNIST/CIFAR experiment. The final test accuracy matters more than the "faster convergence in terms of runtime". The "convergence" in the context of neural networks can be easily influenced by learning rate schedules, and the test accuracy at the beginning of the run does not tell much about the final performance.

**Methods And Evaluation Criteria:**

The method and evaluation make sense.

**Other Comments Or Suggestions:**

1. Is the "non-standard linear algebra operations" a well-known terminology? SVD and QR decompositions feel quite standard.
2. In Sec 6.4, a six-layer network is used. The first four layers are column orthonormal, and the output layer is unconstrained. How about the missing layer?

**Other Strengths And Weaknesses:**

The paper is well-written, and the presentation is clear. The idea is straightforward, and the results look competitive compared to baselines. The weakness is the lack of large-scale practical applications. All the experiments are either toy problems or applications on tiny-scale neural networks.

**Questions For Authors:**

1. What is the number of inner iterations used in the experiments?
2. What is the reasonable range for choosing $r$, especially, how low could the ratio $r/p$ be?
3. Since the random matrix $P_k$ changes every iteration, the momentum for matrix Y does not make sense, correct? I think the inability to use momentum in RSDM is a drawback compared to RGD. Furthermore, it will be more interesting to add RGD with momentum to the baselines in the experiments.

**Relation To Broader Scientific Literature:**

The paper relates to the geometric optimization algorithms on matrix manifolds.

**Theoretical Claims:**

I did not check the proofs of the theorems.

---

> ### Author Rebuttal · Authors · 2025-03-28
>
> We sincerely appreciate the reviewer's thoughtful feedback and overall positive assessment on our work. We would like to take the chance to address the comments in detail.
>
> **1. On the convergence of MNIST/CIFAR experiment.**
>
> **R1**: In our MNIST/CIFAR experiments, our aim was to highlight the efficiency of the proposed method in terms of convergence speed, which aligns with our goal of improving optimization efficiency under orthogonality constraints. Faster convergence is particularly relevant in scenarios where reaching a certain level of accuracy is sufficient or where computational resources are limited.
>
> To ensure a fair comparison, we used a fixed learning rate schedule and tuned both RSDM and RGD for their best performance. This setup is consistent with our theoretical developments, and the experiments serve to demonstrate the practical potential of RSDM in neural network training. We agree that exploring different learning rate schedules is valuable and leave this for future work.
>
> To further evaluate the performance of RSDM in neural network training, we have now added experiment on training a *Vision Transformer*, where we observe that our method consistently achieves higher test accuracy throughout the entire training process (rather than only in early iterations). Please see **R2** for more details.
>
> **2. Lack of large-scale experiments.**
>
> **R2**: Thank you for the suggestion. We have added an additional experiment on training *(orthogonal) Vision Transformer*. Following [1], we impose orthogonality constraint on the query, key and value projection matrices. We train a 6-layer, 4-head transformer with embedding dimension 1024 and 64 patches, on CIFAR10. This scales the number of parameters from 5.4M in Section 6.4 to *28.5M*. We set $r = 300$ for RSDM and tune the stepsize for both RSDM and RGD. The experiment results are included on https://anonymous.4open.science/r/ICML2025-additional-experiments-08E5. We observe that RSDM converges faster than RGD in test accuracy with a non-negligible gap, throughout training process. This demonstrates the potential of RSDM to improve large-scale model training.
>
> As our main focus is on the theoretical development of the framework, testing more diverse applications is beyond the scope of this paper. We plan to explore this direction in the future.
>
> **3. On the terminology of "non-standard linear algebra operations".**
>
> **R3**: Thank you for the question. In this paper, "non-standard linear algebra operations" refer to operations such as matrix decomposition, matrix inverse and matrix exponential. These operations generally have significantly higher computational complexity compared to standard operations like matrix multiplication or elementwise operations. We will clarify the definition in our revised version.
>
> **4. On the orthonormal layers in Sec 6.4.**
>
> **R4**: This is a typo. All the first five layers are orthonormal.
>
> **5. The number of inner iterations used in the experiments.**
>
> **R5**: In the experiments, we implement RSDM according to Algorithm 1, which does not have inner iterations.
>
>
> **6. Reasonable range for $r$.**
>
> **R6**: The suitable range of $r$ depends on the problems. As shown in Figure 3, RSDM demonstrates improved convergence over RGD across a wide range of $r$ values. Determining the optimal $r$ is indeed an important and valuable direction for future work.
>
> **7. On the RSDM with momentum.**
>
> **R7**: Thank you for the suggestion. The idea of combining RSDM with momentum is indeed worth exploring. There are several feasible solutions. One strategy is that we fix $P_k$ for several iterations where we apply momentum. This is equivalent to taking multiple gradient descent (with momentum) for minimizing $\widetilde F_k(Y)$, initialized from $I_r$. To test the viability of the proposed strategy, we evaluate RSDM with momentum and compare against RGD with momentum on the PCA problem. The result is included on https://anonymous.4open.science/r/ICML2025-additional-experiments-08E5. We see that the RSDM with momentum improves the convergence of RSDM, which supports the potential of incorporating momentum into our framework.
>
> We hope our responses have addressed your comments. If there are any further questions or suggestions, we would be happy to address them.

---

### Official Review · Reviewer_ns8p · 2025-03-16

**Overall Recommendation:** 4

**Summary:**

The paper presents an efficient Riemannian optimizer for Stiefel manifolds, introducing two parameterization strategies that reduce the computational complexity of optimization steps while ensuring rigorous convergence analysis.

**Claims And Evidence:**

The paper introduces two parameterization strategies—orthogonal and permutation sampling—applied in both deterministic and stochastic settings. It establishes that Riemannian Stochastic Differential Manifolds (RSDM) achieve the same convergence rate as Riemannian Gradient Descent (RGD) when $p \geq Cn$ and demonstrate significantly higher efficiency than RGD when $p \leq n$. Additionally, the paper provides a detailed analysis of the trade-offs between convergence and efficiency across different parameterization methods. These claims are substantiated by rigorous theoretical results and empirical experiments.

**Essential References Not Discussed:**

The paper overlooks several relevant orthogonal and Riemannian optimization methods, including:

1. **Cheap Orthogonal Constraints in Neural Networks: A Simple Parametrization of the Orthogonal and Unitary Group**
2. **Siamese Networks: The Tale of Two Manifolds**

Incorporating a discussion of these works could provide a more comprehensive comparison and contextualize the proposed approach within the broader landscape of orthogonal optimization techniques.

**Experimental Designs Or Analyses:**

I have checked the experimental designs and found no significant issues.

**Methods And Evaluation Criteria:**

The evaluation is well-aligned with the paper's contributions and application. Notably, the theoretical results are thoroughly developed and rigorously justified. While the experiments are conducted on a limited set of datasets and architectures, this is a minor limitation given the paper’s primary focus on theoretical advancements.

**Other Comments Or Suggestions:**

The paper would be significantly strengthened by incorporating more diverse experiments on challenging tasks. However, given its primary focus on theoretical contributions, this limitation appears to be a minor concern.

**Other Strengths And Weaknesses:**

Several important methods for optimization with orthogonal constraints are not discussed or compared with the proposed approach. Additionally, the experiments are conducted on a very limited set of datasets and architectures. To strengthen the evaluation, it is recommended to test the proposed method on more recent architectures, such as Vision Transformers, and more challenging datasets, such as the Fine-Grained Visual Categorization (FGVC) datasets [3]. Furthermore, the practical benefits of the proposed parameterization are not clearly demonstrated in real-world applications that require orthogonal constraints, limiting its applicability beyond theoretical analysis.

**References:**

[3] Zhang, Y., Tang, H., & Jia, K. (2018). Fine-Grained Visual Categorization using Meta-Learning Optimization with Sample Selection of Auxiliary Data. *arXiv preprint arXiv:1807.10916.*

**Questions For Authors:**

I have no questions for the authors.

**Relation To Broader Scientific Literature:**

While the paper introduces a Riemannian optimizer for the Stiefel manifold, its underlying ideas have the potential to extend to general Riemannian manifolds or, at the very least, a broader class of quotient manifolds.

**Theoretical Claims:**

I have carefully reviewed all the theoretical proofs and found no major issues.

---

> ### Author Rebuttal · Authors · 2025-03-28
>
> We sincerely thank the reviewer for the insightful comments and feedback. We would like to address your comments as follows.
>
>
>
> **1. Discussions on relevant references.**
>
> **R1**: We thank the reviewer for highlighting references [1,2]. Reference [1] re-parameterizes variables in Euclidean space via the Lie exponential map. However, this approach requires differentiating through the exponential map, which can be computationally expensive. Moreover, the re-parameterization may alter the loss landscape, making convergence analysis more difficult. In contrast, our method comes with convergence guarantees, which [1] does not provide.
>
> Reference [2] formulates the problem of training a Siamese network as an optimization problem on the Stiefel manifold and employs Riemannian (stochastic) gradient descent with retraction for parameter updates. Our proposed algorithm is task-agnostic, so it can be applied to improve the optimization in [2] as well.
>
> We will include the above discussions in the revised manuscript.
>
>
> [1] Cheap Orthogonal Constraints in Neural Networks: A Simple Parametrization of the Orthogonal and Unitary Group
>
> [2] Siamese Networks: The Tale of Two Manifolds.
>
>
>
> **2. More practical and diverse datasets and tasks.**
>
> **R2**: We thank the reviewer for the comment. As noted by the reviewer, our primary focus is on theoretical developments, and we have followed prior works in using standard benchmarks. As suggested by the reviewer, we have now conducted additional experiments on *(orthogonal) vision transformer*. We followed [1] by imposing orthogonality constraint on the query, key and value projection matrices. We trained a 6-layer, 4-head transformer (embedding dimension 1024, 64 patches) on CIFAR10. We set $r = 300$ for RSDM and tune the stepsize for both RSDM and RGD. The experiment results are included on  https://anonymous.4open.science/r/ICML2025-additional-experiments-08E5. We observe our proposed RSDM converges faster than RGD in both test accuracy and training loss. This validates the potential of RSDM in more practical settings. We plan to extend the evaluation to additional real-world tasks and datasets in future work.
>
> [1] Fei et al. O-vit: Orthogonal vision transformer. arXiv:2201.12133.
>
>
> We hope we have addressed your concerns. We would be happy to respond to any remaining questions you may have. Once again, we appreciate the reviewer's time and valuable comments.

---

> > ### Comment · Reviewer_ns8p · 2025-04-04
> >
> > Most of my concerns were addressed in this rebuttal as well as the replies in other reviewers. Hence, I decided to adjust my rating accordingly.

---

> > > ### Author Response · Authors · 2025-04-04
> > >
> > > Dear Reviewer ns8p
> > >
> > > We are deeply encouraged that our responses have addressed your concerns, and we sincerely appreciate your decision to raise the score to 4. Thank you again for your valuable time and effort in reviewing our paper.
> > >
> > > Best regards
> > >
> > > Authors

---

### Official Review · Reviewer_gbXw · 2025-03-16

**Overall Recommendation:** 3

**Summary:**

This paper develops a new approach to performing optimization with orthogonality constraints, with an emphasis on keeping computational complexity low. The authors are able to provide convergence bounds in expectation for nonconvex losses. The authors then run a number of experiments on well-known baselines to confirm the utility of their method.

**Claims And Evidence:**

Yes.

**Essential References Not Discussed:**

No.

**Experimental Designs Or Analyses:**

N/A.

**Methods And Evaluation Criteria:**

Yes.

**Other Comments Or Suggestions:**

On the x-axes of Figures 3,4, and 5, please add a label. In this wall-clock time? Number of optimization steps? This information is helpful to know, as most deep learning researchers have a sense of how long it takes to train e.g., MNIST, and it is important to understand the slowdown (if there is one) your algorithm induces.

**Other Strengths And Weaknesses:**

The paper is nicely written, in a way that I (a non-expert in this area) could follow along and understand what was being done. I worry that this work might be too incremental, after having looked through some of the references provided in the introduction. But as I state above, I am not particularly familiar with this literature so I could be wrong about the novelty of this work.

**Questions For Authors:**

No.

**Relation To Broader Scientific Literature:**

This would be interesting to anyone who requires optimizing with orthogonality constraints in their research.

**Theoretical Claims:**

I did not go through the main proofs in detail, but the convergence results in 5.1.1 and 5.1.2 make sense, and are well-explained by the authors.

---

> ### Author Rebuttal · Authors · 2025-03-28
>
> We are grateful that the reviewer found our work interesting and well-written. Below are our responses to your comments.
>
>
> **1. On the novelty of this work compared to related works.**
>
> **R1**: Thank you for the comment. We would like to clarify the novelty of our work. We have carefully compared with related works both theoretically and numerically in the main paper and in Appendix I, J. Our method is well-motivated from Riemannian geometry, which gives a principled foundation for the design and convergence analysis of the algorithms within the framework of Riemannian optimization. This geometric perspective also allows our method to generalize naturally to more complex manifolds, including quotient manifolds, as shown in Section 5.4 and Appendix C.  We believe this provides both theoretical and practical value. We will expand the discussion on the novelty and contributions of our work in the revised manuscript to make this clearer.
>
>
> **2. Add label to x-axes of Figure 3, 4, 5.**
>
> **R2**: Thank you for the question. We have already included the x-axes label for Figure 3,4,5, which is (wall-clock) Time.
>
> We hope we have addressed all your comments and concerns. If you have further questions, we would be happy to address them.

---

### Decision · Program_Chairs · 2025-05-01

**Decision:**

Accept (poster)

**Comment:**

This paper proposes a randomized Riemannian submanifold method for optimization on the Stiefel manifold and studies its convergence behavior under various assumptions on the objective function. While the reviewers find merits in the contribution, they find the numerical experiments somewhat limited. Also, there is no discussion on what applications will give rise to functions that satisfy the Riemannian PL condition, and so the relevance of the condition is unclear. Lastly, as suggested by a reviewer's comment, the term "non-standard linear algebra operations" is inappropriate. The examples cited by the author(s), i.e., matrix decomposition and matrix inverse, are by most account standard. Moreover, the computation of the matrix exponential has been extensively studied for decades; see, e.g., Higham: The Scaling and Squaring Method for the Matrix Exponential Revisited. SIMAX 26, 2005 and the references therein. Thus, the author(s) should simply state that the linear algebra operations they use have lower complexities than those used in other works, if that is indeed the case. Overall, the author(s) should take into account the reviewers' comments when they revise the paper.